# DIFFERENTIABLE CAUSAL DISCOVERY FOR LATENT HIERARCHICAL CAUSAL MODELS

**Parjanya Prajakta Prashant**[1]    **Ignavier Ng**[2]    **Kun Zhang**[2,3]    **Biwei Huang**[1]

[1]University of California San Diego
[2]Carnegie Mellon University
[3]Mohamed bin Zayed University of Artificial Intelligence

## ABSTRACT

Discovering causal structures with latent variables from observational data is a fundamental challenge in causal discovery. Existing methods often rely on constraint-based, iterative discrete searches, limiting their scalability for large numbers of variables. Moreover, these methods frequently assume linearity or invertibility, restricting their applicability to real-world scenarios. We present new theoretical results on the identifiability of non-linear latent hierarchical causal models, relaxing previous assumptions in the literature about the deterministic nature of latent variables and exogenous noise. Building on these insights, we develop a novel differentiable causal discovery algorithm that efficiently estimates the structure of such models. To the best of our knowledge, this is the first work to propose a differentiable causal discovery method for non-linear latent hierarchical models. Our approach outperforms existing methods in both accuracy and scalability. Furthermore, we demonstrate its practical utility by learning interpretable hierarchical latent structures from high-dimensional image data and demonstrate its effectiveness on downstream tasks such as transfer learning.

## 1 INTRODUCTION

Causal discovery, the task of inferring causal relationships from observational data, is fundamental to understanding complex systems across various scientific disciplines. Traditional causal discovery methods often assume the absence of latent confounders, a property known as causal sufficiency (Spirtes et al., 2001; Chickering, 2002). However, this condition is frequently violated in real-world scenarios, where latent variables can introduce spurious correlations among observed variables. For instance, in image analysis, latent semantic variables often act as common causes for multiple pixels, creating complex dependencies that are not directly observable.

Recognizing the limitations of causal sufficiency conditions, researchers have developed various approaches to handle latent confounders. Fast Causal Inference (FCI) and its extensions (Spirtes et al., 2001; Pearl et al., 2000; Zhang, 2008; Colombo et al., 2012; Claassen et al., 2013; Akbari et al., 2021) leverage conditional independence information to infer a class of possible causal graphs. While these methods can identify the presence of latent confounders, they do not provide information about causal relationships among the latent variables themselves.

Therefore, another line of research has emerged, focusing on methods that can identify causal relations between latent variables. These approaches typically introduce additional parametric constraints, such as linearity or discrete data, to make the problem tractable. Notable examples include methods based on rank constraints (Silva et al., 2006; Kummerfeld & Ramsey, 2016; Huang et al., 2022; Xie et al., 2022; Dong et al., 2023), higher-order moments (Shimizu et al., 2009; Xie et al., 2020; Adams et al., 2021; Chen et al., 2022), copula models (Cui et al., 2018), multiple domains based methods (Zeng et al., 2021; Li et al., 2023; Sturma et al., 2024), matrix-decomposition (Anandkumar et al., 2013) and mixture models (Kivva et al., 2021).

Recently, Kong et al. (2023) proposed a method for identifying the causal structure without assuming linearity for a broad class of structures called latent hierarchical models. These models appear across various domains, including gene regulatory networks (Gitter et al., 2016), image data (Higgins et al.,

2017), political science (Weinstein & Blei, 2024), and epidemiology (O'Brien et al., 2019). Latent hierarchical models allow latent variables with no observed children and multiple paths between two variables. However, Kong et al. (2023) assume that latent variables and exogenous noise are deterministic functions of measured variables, limiting the applicability of their identifiability results. In this paper, we establish the identifiability of non-linear latent hierarchical models without this condition. To the best of our knowledge, we are the first to prove identifiability for non-linear hierarchical latent models under general conditions.

Moreover, these methods often employ an iterative procedure, learning local graph structures sequentially. Such iterative approaches often face challenges including scalability issues (Chickering et al., 2004; Niu et al., 2024), error propagation, and sensitivity to testing order (Spirtes, 2010; Colombo et al., 2012). To address these empirical issues, researchers have proposed differentiable causal discovery methods (Zheng et al., 2018; Yu et al., 2019; Zhang et al., 2019; Zheng et al., 2020; Sethuraman et al., 2023; Nazaret et al., 2023). Unlike iterative methods or discrete search-based approaches, these methods formulate the combinatorial search problem of causal discovery as a continuous optimization algorithm, incorporating differentiable algebraic constraints to enforce structural requirements, such as acyclicity. However, these methods typically assume no latent variables. Recent work has extended these methods to include latent variables, but these approaches are limited by their linear assumptions and inability to recover causal relations between latent variables (Bhattacharya et al., 2021; Ma et al., 2024; Ng et al., 2024a). To tackle these limitations, we propose a scalable differentiable causal discovery method for non-linear latent hierarchical models.

Our key contributions are as follows:

1. We establish theoretical guarantees for the identifiability of non-linear latent hierarchical models under considerably relaxed conditions. Notably, we eliminate the requirement for latent variables and exogenous noise to be deterministic functions of measured variables.

2. We introduce a novel differentiable causal discovery method for latent hierarchical models. Through comprehensive experiments on synthetic datasets, we demonstrate our method's improved performance and scalability compared to existing approaches for latent hierarchical models. We also showcase our method's real-world applicability by learning a hierarchical latent model for high-dimensional image data.

## 2 RELATED WORK

**Causal Discovery for Latent Hierarchical Models:** Prior work extensively focuses on the linear case. Silva et al. (2006) and Kummerfeld & Ramsey (2016) utilize tetrad conditions—the rank of each $2 \times 2$ sub-covariance matrix—to discover latent variables. However, they require further structural conditions, such as trees and three measured pure children for each latent variable. Anandkumar et al. (2013) employ matrix factorization to identify latent variables based on decomposing the covariance matrix. Xie et al. (2020; 2022) use an independent noise condition for linear latent models with non-Gaussian noise. Huang et al. (2022) use rank deficiency constraints on pairs of measured variable sets to identify the minimum number of latent variables that d-separate the two sets. Dong et al. (2023) extend this framework to allow measured variables to be parents of other variables as well. Both these methods use an iterative search procedure with rank deficiency test to discover the latent graph, with a time complexity at least quadratic in the number of variables. Ng et al. (2024a) develop a score-based search procedure based on the structural conditions considered by Silva et al. (2006); Huang et al. (2022).

Kong et al. (2023) introduce identifiability results for the non-linear case when the latent variables and exogenous noise are a differentiable invertible function of the measured variables ($z, \epsilon = f(x)$). Similar to other methods, they rely on an iterative search procedure that requires training at least $\mathcal{O}(ln^2)$ generative models, where $n$ is the number of measured variables and $l$ is the number of layers in the model. Welch et al. (2025) relax structural constraints for non-linear latent models where the exogenous noise is gaussian and the causal relations between the latent and observed variables are linear.

Another line of work focuses on learning hierarchical structures for discrete latent variables (Pearl, 1988; Choi et al., 2011; Gu & Dunson, 2023). However, these approaches assume the measured variables are discrete, which often does not hold in many real-world scenarios, such as images.

Kong et al. (2024) allow continuous variables to be adjacent to latent discrete variables. However, causal relationships and the hierarchical structure are only learned for discrete variables. Moreover, latent variables are assumed to be a deterministic invertible function of the measured variables, similar to Kong et al. (2023).

**Differentiable Causal Discovery:** NOTEARS introduces a continuous optimization-based algorithm to learn linear directed acyclic graphs (DAG) by reformulating graphical constraints into differentiable ones (Zheng et al., 2018). Subsequent work parametrizes DAGs using neural networks (Yu et al., 2019; Zhang et al., 2019; Ng et al., 2022). Zheng et al. (2020) extended these approaches to non-parametric DAGs. While these approaches scale well, they usually assume the absence of any latent variables in the DAG.

Recent differentiable causal discovery algorithms address the existence of latent variables (Bhattacharya et al., 2021; Bellot & van der Schaar, 2021). However, these approaches only focus on the causal relationships among observed variables, fail to capture causal relationships involving latent variables, and often assume linearity in the causal structure. Ma et al. (2024) build on similar assumptions but employ a supervised learning framework for causal discovery. Despite these advances, there are concerns that some of these methods may not truly perform causal discovery (Reisach et al., 2021; Seng et al., 2024). To address these issues, Ng et al. (2024b) advocate for improved evaluation practices such as non-equal noise variances in the data generation process.

**Causal Representation Learning:** Causal representation learning (Schölkopf et al., 2021) is closely related to latent causal discovery. Non-linear Independent Component Analysis methods aim to recover independent latent sources from a set of measured variables. However, they do not consider generic dependence between latent variables and rely on additional assumptions such as existence of auxiliary variables (Hyvarinen et al., 2019; Hyvärinen et al., 2023) or restrictions on the mixing function (Zheng et al., 2022; Buchholz et al., 2022). Some methods attempt to model dependencies among latent variables explicitly. For example, He et al. (2018) propose using a graphical structure to represent these dependencies, though their approach lacks identifiability guarantees. Yang et al. (2021) assume linear relations and incorporate additional supervision information to establish the identifiability. Others utilize interventional or multi-domain data to learn these dependencies (Brehmer et al., 2022; Subramanian et al., 2022; Squires et al., 2023; von Kügelgen et al., 2023; Zhang et al., 2024).

## 3 NON-LINEAR LATENT HIERARCHICAL CAUSAL MODELS

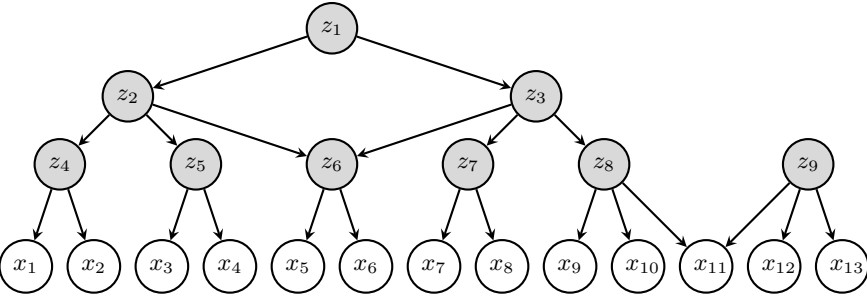

Figure 1: Example of a graph we consider. Note that we allow multiple paths between two nodes and hence generalize trees. The latent variables are shaded.

We consider a latent hierarchical causal model represented by a directed acyclic graph (DAG) $\mathcal{G} = (\mathcal{V}, \mathcal{E})$, where $\mathcal{V} = \mathbb{Z} \cup \mathbb{X}$ comprises latent variables $\mathbb{Z} = \{z_1, z_2, \ldots, z_{n_z}\}$ and measured variables $\mathbb{X} = \{x_1, x_2, \ldots, x_{n_x}\}$, and $\mathcal{E}$ denotes the set of edges representing causal relationships. The variables follow the data-generating procedure:

$$
\begin{aligned}
z_j &= f_{z_j}(\mathrm{Pa}(z_j), \varepsilon_{z_j}), \\
x_i &= f_{x_i}(\mathrm{Pa}(x_i), \varepsilon_{x_i})
\end{aligned}
\tag{1}
$$

where $\mathrm{Pa}(\cdot)$ represents the set of parent variables of a given node in $\mathcal{G}$, and $\mathrm{Pa}(x_i), \mathrm{Pa}(z_j) \subseteq \mathbb{Z}$.

The structure of DAG $\mathcal{G}$ can be characterized by binary matrices $\boldsymbol{M}^z$ and $\boldsymbol{M}^x$, where $\boldsymbol{M}^z_{ij} = 1$ if and only if there is an edge $z_i \to z_j$, and $\boldsymbol{M}^x_{ij} = 1$ if and only if there is an edge $z_i \to x_j$. Without loss of generality, we assume $\boldsymbol{M}^z$ is upper triangular. The binary adjacency matrix $\boldsymbol{M}$ is obtained by horizontally concatenating $\boldsymbol{M}^z$ and $\boldsymbol{M}^x$, i.e., $\boldsymbol{M} = [\boldsymbol{M}^z \mid \boldsymbol{M}^x]$.

The goal of this work is to recover the binary matrix $\boldsymbol{M}$ which characterizes the structure of DAG $\mathcal{G}$. Since the labeling of latent variables in general cannot be identified, we aim to recover $\boldsymbol{M}$ upto the relabeling of the latent variables.

In general, latent hierarchical models are not identifiable. For example, consider the model shown in Figure 7a. It is in general not possible to disentangle the effects of $z_1, z_2, z_3$ and identify the structure or their values. Hence, we require structural conditions on the model to make it identifiable. Prior work has addressed this challenge through various constraints. Silva et al. (2006); Kummerfeld & Ramsey (2016); Huang et al. (2022); Kong et al. (2023) propose requirements on the minimum number of measured pure children, allowing latent variables to leave sufficient footprints in the measured variables. Additionally, researchers have often assumed tree-like structures to prove identifiability of hierarchical models (Choi et al., 2011; Drton et al., 2017; Huang et al., 2022; Kong et al., 2023).

In order for our model to be identifiable, we consider the following structural conditions:

**Definition 1** (Pure Children). *$v_i$ is a pure child of another variable $v_j$, if $v_j$ is the only parent of $v_i$ in the graph, i.e., $Pa(v_i) = \{v_j\}$.*

**Condition 1.** *(i) Each latent variable has at least two pure children. (ii) For any latent variable $z_i \in \mathbb{Z}$, let $\mathcal{D}_i = De(z_i) \cap \mathbb{X}$ be the set of measured descendants of $z_i$ where $De(\cdot)$ denotes the descendants. Then, for all $x_j, x_k \in \mathcal{D}_i$, $d(z_i, x_j) = d(z_i, x_k)$ where $d(\cdot, \cdot)$ denotes the length of the directed path between two nodes in the graph $\mathcal{G}$.*

We provide additional discussion on the above condition in Appendix C.

Define $\mathbb{Z}^l = \{z_i \in \mathbb{Z} : \forall x_j \in De(z_i) \cap \mathbb{X}, d(z_i, x_j) = l\}$. This denotes the set of latent variables in the $l^{\text{th}}$ layer of the model. Henceforth, we denote the vector obtained by concatenating the elements in $\mathbb{Z}^l$ as $\boldsymbol{z}^l$ and $z^l_i$ as the $i^{\text{th}}$ element of layer $l$.

Note that since any node in $\mathbb{Z}^i$ has parents only in $\mathbb{Z}^{i-1}$, the adjacency matrix $\boldsymbol{M}$ can be transformed via suitable column and row permutations to the block upper-triangular structure as shown below:

$$\boldsymbol{M} = \begin{bmatrix} \boldsymbol{0} & \boldsymbol{M}^k & \boldsymbol{0} & \cdots & \boldsymbol{0} \\ \boldsymbol{0} & \boldsymbol{0} & \boldsymbol{M}^{k-1} & \cdots & \boldsymbol{0} \\ \vdots & \vdots & \vdots & \ddots & \vdots \\ \boldsymbol{0} & \boldsymbol{0} & \boldsymbol{0} & \cdots & \boldsymbol{M}^1 \end{bmatrix} \tag{2}$$

where $\boldsymbol{M}^i \in \{0, 1\}^{|\mathbb{Z}^i| \times |\mathbb{Z}^{i-1}|}$ are binary matrices that model the causal structure between $\mathbb{Z}^i$ and $\mathbb{Z}^{i-1}$. $\boldsymbol{M}^1 \in \{0, 1\}^{|\mathbb{Z}^1| \times |\mathbb{X}|}$ is the binary matrix between $\mathbb{Z}^1$ and $\mathbb{X}$. Henceforth in the paper, we assume $\boldsymbol{M}$ is modeled this way and hence always satisfies condition 1 (ii).

Our structural conditions are fairly general. We reduce the required number of measured variables relative to Silva et al. (2006) and Kummerfeld & Ramsey (2016). Unlike Choi et al. (2011) and Drton et al. (2017), we do not restrict children to have only one latent parent. Furthermore, our method imposes no constraints on the neighborhood structure of variables as in Huang et al. (2022); Xie et al. (2022).

**Additional Notation:** For any matrix $A$, we use $A_{i,:}$ to denote its $i^{th}$ row, $A_{:,j}$ for its $j^{th}$ column, and $A_{i,j}$ for the element at the $i^{th}$ row and $j^{th}$ column. For a set $\mathbb{A}$, $\boldsymbol{a}$ denotes the vector obtained by concatenating all the elements in $\mathbb{A}$ and $\boldsymbol{a}_i$ denotes the $i^{th}$ element of $\boldsymbol{a}$.

## 4 IDENTIFIABILITY THEORY

In this section, we describe the identifiability theory for general latent hierarchical models. Prior work has used rank constraints on the observed distribution as a graphical indicator of latent variables (Silva et al., 2006; Huang et al., 2022; Dong et al., 2023). They use the rank of the cross-

covariance matrix between two sets of measured variables to determine the number of latent variables that d-separate the two measured sets. However, this criterion only works for linear cases hence limiting the identifiability results. We propose a novel indicator which allows us to determine the number of latent variables which d-separate the two measured sets in the general case.

**Intuition:** Consider the case where a set of latent variables, denoted by $\mathbb{Z}$, d-separates two sets of measured variables, $\mathbb{X}$ and $\mathbb{Y}$. In this scenario, the conditional distribution $p(\boldsymbol{y}|\boldsymbol{x})$ can be expressed as: $p(\boldsymbol{y}|\boldsymbol{x}) = \int p(\boldsymbol{y}|\boldsymbol{z})p(\boldsymbol{z}|\boldsymbol{x})d\boldsymbol{z}$. If the cardinality of $\mathbb{Z}$ is smaller than that of $\mathbb{X}$ or $\mathbb{Y}$, this imposes a constraint on the measured distribution. In most cases, the size of $\mathbb{Z}$ would be equal to the minimum dimension of $\boldsymbol{z}$ such that $p(\boldsymbol{y}|\boldsymbol{x})$ can be written as $\int p(\boldsymbol{y}|\boldsymbol{z})p(\boldsymbol{z}|\boldsymbol{x})d\boldsymbol{z}$. Based on this observation, we formulate a criterion based on the rank of the Jacobian of the function $\mathbb{E}[\boldsymbol{y}|\boldsymbol{x}]$, which allows us to show that the general case holds with probability one.

In order to rigorously prove identifiability, we introduce some standard differentiability and faithfulness conditions (Huang et al., 2022; Dong et al., 2023).

**Condition 2** (Generalized Faithfulness). *A probability distribution $P$ is faithful to a DAG $\mathcal{G}$ if every rank Jacobian constraint on a pair of set of measured variables that holds in $P$ is entailed by every structural equation model with respect to $\mathcal{G}$.*

Faithfulness conditions are widely used in causal discovery (Spirtes et al., 2001; Zhang, 2008; Silva et al., 2006; Huang et al., 2022). This is often justified by the fact that the Lebesgue measure of distributions violating faithfulness has been shown to be zero. The following proposition justifies the faithfulness condition for non-linear latent hierarchical models along similar lines.

**Proposition 1.** *The probability of a distribution $P$ generated by a structural model with respect to $\mathcal{G}$ violating Generalized Faithfulness is zero.*

**Condition 3** (Differentiability). *(i) For every pair of measured sets $\mathbb{X}$ and $\mathbb{Y}$, the function $f : \mathbb{R}^{|\mathbb{X}|} \to \mathbb{R}^{|\mathbb{Y}|}$ defined as $f(\boldsymbol{x}) = \mathbb{E}[\boldsymbol{y}|\boldsymbol{x}]$ is continuously differentiable. (ii) For every pair of measured set $\mathbb{X}$ and latent set $\mathbb{Z}$, there exists a continuous differentiable function $g : \mathbb{R}^{|\mathbb{X}|} \to \mathbb{R}^{|\mathbb{Z}|}$ such that $p(\boldsymbol{z}|\boldsymbol{x}) = p(\boldsymbol{z}|g(\boldsymbol{x}))$.*

Our approach considerably relaxes the constraints compared to existing work. Unlike previous methods that require linear relationships (Huang et al., 2022; Dong et al., 2023) or deterministic functions $(z, \epsilon = f(x))$ (Kong et al., 2023), our framework accommodates a broader class of non-linear relationships between variables. Also, note that these conditions are sufficient but not necessary. In Section 6 we show we can identify structures even when Condition 3 does not hold.

We introduce a theorem that relates the graphical structure to a constraint of the distribution between two sets of measured variables.

**Theorem 1.** *Let $\mathcal{G}$ be a hierarchical latent causal model satisfying Condition 1. For any two sets of measured variables $\mathbb{X}$ and $\mathbb{Y}$ in $\mathcal{G}$, let $f(\boldsymbol{x}) = \mathbb{E}[\boldsymbol{y}|\boldsymbol{x}]$. Under the faithfulness and differentiability conditions, for any $r < |\mathbb{X}|, |\mathbb{Y}|$, the rank of the Jacobian matrix $\mathbf{J}_f = \frac{\partial f}{\partial \boldsymbol{x}} = r$ if and only if the size of the smallest set of latent variables that d-separates $\mathbb{X}$ from $\mathbb{Y}$ is $r$. Formally,*

$$rank(\mathbf{J}_f) = \min_{\mathbb{Z}} |\mathbb{Z}| \quad such \ that \quad \mathbb{X} \perp\!\!\!\perp_{\mathcal{G}} \mathbb{Y}|\mathbb{Z} \tag{3}$$

*where $\mathbb{Z}$ is a subset of latent variables in $\mathcal{G}$, and $\perp\!\!\!\perp_{\mathcal{G}}$ denotes d-separation in the graph $\mathcal{G}$.*

Henceforth, we use $r(\mathbb{X}, \mathbb{Y})$ to denote the rank of the Jacobian of the function $\mathbb{E}[\boldsymbol{y}|\boldsymbol{x}]$. Moreover, it can be shown that pure descendants of latent variables can be used as a surrogate to calculate d-separation between sets of latent variables as stated in Theorem 2 below. This theorem is partly inspired by Huang et al. (2022).

**Theorem 2.** *Let $\mathcal{G}$ be a hierarchical latent causal model satisfying Condition 1. Let $\mathbb{Z}_X, \mathbb{Z}_Y \subseteq \mathbb{Z}^i$ be two disjoint subsets of latent variables in some layer $\mathbb{Z}^i$ in $\mathcal{G}$, i.e., $\mathbb{Z}_X \cap \mathbb{Z}_Y = \emptyset$. Let $\mathbb{X}$ be the set of measured variables that are d-separated by $\mathbb{Z}_X$ from all other measured variables in $\mathcal{G}$ and let $\mathbb{Y}$ be the set of measured variables that are d-separated by $\mathbb{Z}_Y$ from all other measured variables in $\mathcal{G}$ such that $\mathbb{X} \cap \mathbb{Y} = \emptyset$. Then,*

$$r(\mathbb{Z}_X, \mathbb{Z}_Y) = r(\mathbb{X}, \mathbb{Y})$$

The two theorems presented above establish a crucial link between the measured distribution and the underlying graph structure. Building upon this foundation, we now introduce three lemmas that are instrumental in proving identifiability. These lemmas provide a systematic approach to uncover the latent structure:

**Lemma 1.** *Let $\mathcal{G}$ be a graph satisfying Conditions 1. A set of measured variables $\mathbb{S}$ are pure children of the same parent if and only if for any subset $\mathbb{T} \subseteq \mathbb{S}$, $r(\mathbb{T}, \mathbb{X} \setminus \mathbb{T}) = 1$.*

**Lemma 2.** *Let $\mathcal{G}$ be a hierarchical latent causal model satisfying Conditions 1. Under the generalized faithfulness condition, for any measured variable $c \in \mathbb{X}$ and any set of latent variables $\mathbb{P} \subseteq \mathbb{Z}^1$, $c$ is a child of exactly the variables in $\mathbb{P}$ if and only if the following conditions hold:*

1. *For each $\mathbb{S} \subseteq \mathbb{X}$ such that $|\mathbb{S} \cap Ch(z_i)| = 1$ for each $z_i \in \mathbb{P}$, where $Ch(z_i)$ denotes the set of pure children of $z_i$:*

$$r(\mathbb{S}, \mathbb{X} \setminus (\mathbb{S} \cup \{c\})) = r(\mathbb{S} \cup \{c\}, \mathbb{X} \setminus (\mathbb{S} \cup \{c\}))$$

2. *The equality in condition (1) does not hold for any proper subset of $\mathbb{P}$.*

**Lemma 3.** *Let $\mathcal{G}$ be a graph satisfying Conditions 1. A measured variable $c$ has no parent if and only if $r(\{c\}, \mathbb{X} \setminus \{c\}) = 0$.*

**Discussion:** Lemma 1 enables us to identify pure children among the measured variables $\mathbb{X}$. For example, in Figure 1 all subsets of $\{x_1, x_2\}$ are d-separated from the rest of the variables by one variable $\{z_4\}$. However, for the set $\{x_1, x_2, x_3, x_4\}$, the subset $\{x_1, x_3\}$ requires two variables $\{z_4, z_5\}$ to be d-separated from the rest of the variables. Lemma 2 provides a method to determine the parents of non-pure children. For example, in Figure 1, consider $\{x_{11}\}$ whose parents are $\{z_8, z_9\}$. For a set like $\{x_9, x_{12}\}$, which contains exactly one pure child of both parents of $\{x_{11}\}$, $r(\{x_9, x_{12}\}, \mathbb{X} \setminus \{x_9, x_{11}, x_{12}\}) = r(\{x_9, x_{12}, x_{11}\}, \mathbb{X} \setminus \{x_9, x_{12}, x_{11}\})$. However, this is not true for any other set of latent variables. Lemma 3 allows us to identify measured variables that have no latent parents.

Having established the necessary lemmas, we now present the identifiability of the graph structure in hierarchical latent causal models.

**Theorem 3.** *Let $\mathcal{G} = (\mathcal{V}, \mathcal{E})$ be a hierarchical latent causal model satisfying Condition 1. Let $\boldsymbol{M}$ be the binary adjacency matrix representing the structure of $\mathcal{G}$. Let data $\mathbb{X}$ be generated according to the structural equation model defined in Equation 1. Given a function $r(\mathbb{S}, \mathbb{T})$ which outputs the minimum number of latent variables that d-separate any two measured sets $\mathbb{S}$ and $\mathbb{T}$, $\boldsymbol{M}$ is identifiable up to the permutation of the latent variables.*

The proof for Theorem 3 leverages the preceding lemmas and recursion. We begin by applying Lemmas 1, 2, and 3 to infer the structure between $\mathbb{Z}^1$ and $\mathbb{X}$. Theorem 2 then allows us to relate d-separation between sets in $\mathbb{Z}^1$ to their pure children in $\mathbb{X}$. Thus, this process can be applied recursively to higher levels of the hierarchy, enabling the identification of the entire graph structure.

## 5 DIFFERENTIABLE CAUSAL DISCOVERY APPROACH

In the previous section, we demonstrated that hierarchical models satisfying Condition 1 yield a unique hierarchical structure for a given distribution of measured variables. To learn the causal structure, two key steps must be performed: (i) matching the observed data distribution, and (ii) enforcing structural constraints on the model. [1]

### 5.1 MATCHING THE DATA DISTRIBUTION

To learn the causal structure, we consider the structural equation models (SEMs) in Equation 1 explicitly parameterized by the binary adjacency matrix $\boldsymbol{M}$.

$$
\begin{aligned}
z_j^i &= f_j^i(\boldsymbol{M}^{i+1} \odot \boldsymbol{z}^{i+1}, \varepsilon_{z_j^i}), \\
x_j &= g_j(\boldsymbol{M}^1 \odot \boldsymbol{z}^1, \varepsilon_{x_j}).
\end{aligned}
\tag{4}
$$

---

[1]Software for implementation: https://github.com/parjanya20/latent-causal-models

We employ a variational autoencoder (VAE) (Kingma, 2013) as a generative model to learn the distribution over the measured variables. Let $\theta$ be the parameters of the VAE, and $\boldsymbol{M}$ represent the binary adjacency matrix controlling the structure of the SEM. We aim to maximize the evidence lower bound (ELBO) to approximate the true data distribution.

$$
\begin{aligned}
\log p(\boldsymbol{x}; \theta, \boldsymbol{M}) &= \log \int p(\boldsymbol{x}|\boldsymbol{\epsilon}; \theta, \boldsymbol{M}) p(\boldsymbol{\epsilon}; \theta) d\boldsymbol{\epsilon} \\
&= \log \int \frac{q(\boldsymbol{\epsilon}|\boldsymbol{x})}{q(\boldsymbol{\epsilon}|\boldsymbol{x})} p(\boldsymbol{x}|\boldsymbol{\epsilon}; \theta, \boldsymbol{M}) p(\boldsymbol{\epsilon}; \theta) d\boldsymbol{\epsilon} \\
&\geq -\text{KL}(q(\boldsymbol{\epsilon}|\boldsymbol{x})\|p(\boldsymbol{\epsilon}; \theta)) + \mathbb{E}_q[\log p(\boldsymbol{x}|\boldsymbol{\epsilon}; \theta, \boldsymbol{M})]
\end{aligned}
\tag{5}
$$

Here, $\boldsymbol{\epsilon}$ represents the latent variable vector obtained by concatenating all individual noise terms $\varepsilon_{z_j^i}$ and $\varepsilon_{x_j}$. The objective of the VAE is to minimize the negative ELBO $\mathcal{L}_{\text{ELBO}}$, where the KL divergence regularizes the variational posterior, and the second term encourages the generative model to match the observed data distribution.

The encoder of the VAE models the approximate posterior $q(\boldsymbol{\epsilon}|\boldsymbol{x})$, mapping the observed data $\boldsymbol{x}$ to the latent space $\boldsymbol{\epsilon}$. An advantage of modeling $q(\boldsymbol{\epsilon}|\boldsymbol{x})$ over $q(\boldsymbol{z}|\boldsymbol{x})$ is that it simplifies the process of enforcing the independence of each dimension of $\boldsymbol{\epsilon}$. The decoder models the conditional likelihood $p(\boldsymbol{x}|\boldsymbol{\epsilon}; \theta, \boldsymbol{M})$, and is designed to follow the SEM equations in Equation 4. This allows the decoder to respect the structural constraints encoded in the binary adjacency matrix $\boldsymbol{M}$, ensuring that the learned distribution also reflects the underlying causal structure.

## 5.2 Enforcing Structural Constraints

In order to enforce structural constraints, we relax the binary adjacency matrix $\boldsymbol{M}$ for gradient-based optimization by using the Gumbel-softmax trick (Jang et al., 2016). Here, $\boldsymbol{M} \sim \sigma(\gamma)$, where $\sigma$ represents the softmax function and $\gamma$ is a trainable parameter representing the logits.

To ensure the causal structure satisfies Condition 1 (ii), we define $\boldsymbol{M}$ as a block upper-triangular matrix as shown in Equation 2. We now introduce the following lemma to justify the constraint required for Condition 1 (i).

**Lemma 4.** *Consider a DAG $\mathcal{G}$ with a binary adjacency matrix $\boldsymbol{M}$. $\mathcal{G}$ satisfies Condition 1 (i) if and only if:*

$$
\left\| \boldsymbol{M}_{i,:} \odot \prod_{j \neq i} (1 - \boldsymbol{M}_{j,:}) \right\|_1 \geq 2 \quad \forall i.
\tag{6}
$$

This lemma ensures that each row in $\boldsymbol{M}$ with descendants must account for at least two pure children, where pure children are those that do not share another parent. The elementwise product with $\prod_{j \neq i}(1 - \boldsymbol{M}_{j,:})$ counts pure children, and the $\ell_1$ norm helps ensure that each row satisfies the minimum number of pure children.

To encourage sparsity and avoid learning spurious edges, we apply an $\ell_1$ regularization on $\sigma(\gamma)$, similar to other differentiable causal discovery methods (Ng et al., 2022; Brouillard et al., 2020).

The optimization objective is formulated as:

$$
\max_{\theta, \gamma} \ \mathbb{E}_{\boldsymbol{M} \sim \sigma(\gamma)} \left[ \text{ELBO}(\theta, \boldsymbol{M}) \right] - \lambda \|\sigma(\gamma)\|_1,
\tag{7}
$$

$$
\text{subject to} \quad \|\boldsymbol{M}_{i,:}\|_1 \left( \left\| \boldsymbol{M}_{i,:} \odot \prod_{j \neq i}(1 - \boldsymbol{M}_{j,:}) \right\|_1 - 2 \right) \geq 0 \quad \forall i.
\tag{8}
$$

Note that we allow some rows of $\boldsymbol{M}$ to go zero. This allows us to learn the number of latent variables. The above method of using Gumbel softmax to approximate the binary adjacency matrices is inspired by Ng et al. (2022); Brouillard et al. (2020).

Furthermore, to ensure the independence of the noise terms $\varepsilon$, we introduce the following independence loss, denoted as $\mathcal{L}_{\text{ind}}(\boldsymbol{\epsilon})$, which minimizes the KL divergence between the joint distribution of $\boldsymbol{\epsilon}$ and the product of individual noise distributions:

$$
\mathcal{L}_{\text{ind}}(\boldsymbol{\epsilon}) = \text{KL}\left( p(\boldsymbol{\epsilon}) \| \prod_j \prod_i p(\varepsilon_{z_j^i}) \prod_j p(\varepsilon_{x_j}) \right).
\tag{9}
$$

This can be estimated using the Donsker-Varadhan representation (Donsker & Varadhan, 1983; Belghazi et al., 2018).

Therefore, the final loss function is:

$$\mathcal{L}_{\text{final}} = -\mathbb{E}_{\boldsymbol{M} \sim \sigma(\gamma)}\left[\text{ELBO}(\theta, \boldsymbol{M})\right] + \lambda_1 \mathcal{L}_{\text{ind}}(\boldsymbol{\epsilon}) + \lambda_2 \|\sigma(\gamma)\|_1$$
$$+ \lambda_3 \Big( \sum_i \max(0, \|\boldsymbol{M}_{i,:}\|_1 (2 - \|\boldsymbol{M}_{i,:} \odot \prod_{j \neq i}(1 - \boldsymbol{M}_{j,:})\|_1)) \Big)^2. \tag{10}$$

## 6 EXPERIMENTS

We conduct empirical studies to examine the efficacy of our differentiable causal discovery method. Specifically, we experiment with synthetic data in Section 6.1 and real image data in Section 6.2.

### 6.1 SYNTHETIC DATA

We conduct experiments on four causal structures given in Figure 4 to validate our method. We consider both trees and v-structures. We compare against other methods designed to discover latent hierarchical causal models, namely KONG (Kong et al., 2023), HUANG (Huang et al., 2022), GIN (Xie et al., 2020) and DeCAMFounder (Agrawal et al., 2023). The structural Hamming distance (SHD) and F1 score are computed for each structure and reported in Table 1. We also report the time taken for each method in seconds. We did not run 1-factor model methods like FOFC (Kummerfeld & Ramsey, 2016) since our data does not meet their conditions, and their implementation results in runtime errors.

For the ground truth graphs, the functions in Equation 1 are modeled using a linear transformation of the input followed by a Tanh or LeakyReLU activation function with $\alpha = 0.2$. The weights for the linear transformation are uniformly sampled from $[-5, -2] \cup [2, 5]$. Exogenous noise is sampled from $[-\alpha, \alpha]$ where $\alpha$ is sampled from $[-3, -1] \cup [1, 3]$. We run three random trials for each graph, and report the mean and standard deviation for each metric. Further details are given in Appendix B.1.

We observe substantial improvement in both the SHD and F1 score compared to the baselines. We note that this improvement is despite the fact that the data does not satisfy Condition 3 since LeakyReLU is not differentiable everywhere. Since other methods are designed for a restrictive class of latent hierarchical models, they are unable to identify the causal graph. Xie et al. (2020) does not predict edges for most of the runs resulting in a mean F1 score close to zero. Agrawal et al. (2023) only discovers edges between observed variables, hence has a high SHD and low F1 score.

The linear baselines (Huang et al., 2022; Xie et al., 2020) are faster than the non-linear methods. This is because they do not have to train a non-linear model like a neural network since all relationships are linear. However, we can see that we require a much shorter runtime compared to Kong et al. (2023) since we only train one neural network instead of $\mathcal{O}(ln^2)$.

### 6.2 IMAGE DATA

In this section, we learn a latent causal graph for the MNIST dataset (LeCun et al., 2010). As shown in Figure 3a, we model a latent hierarchical causal structure followed by a decoder that generates the images. Since our causal discovery approach can be trained end-to-end, incorporating the decoder does not alter our methodology. The convolution decoder allows us to use spatial information and reduce the number of measured variables. We initialize the causal model with three layers and learn the underlying structure. Further training details are provided in Appendix B.2.

For real image data, where the ground truth causal graph is unavailable, we validate our methodology using indirect evaluation techniques. These include visualizing the learned latent features, conducting interventions to interpret the learned representations, and evaluating their transferability across domains and under distribution shifts.

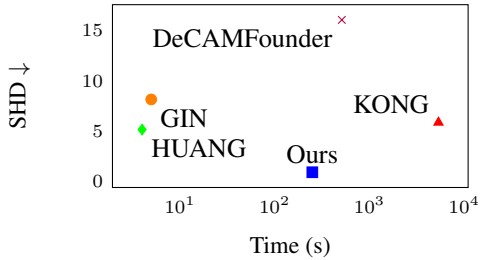
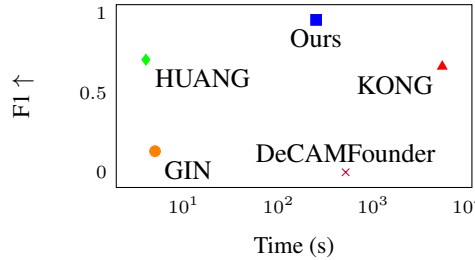

(a) Structural Hamming Distance (SHD) vs. Time. Lower SHD is better.

(b) F1 Score vs. Time. Higher F1 is better.

Figure 2: Performance vs. Time for different causal discovery methods. Time is plotted on a logarithmic scale.

Table 1: Performance of latent hierarchical causal discovery methods on various graphs

|  | Ours | | KONG | | HUANG | | GIN | | DeCAMFounder | |
|---|---|---|---|---|---|---|---|---|---|---|
| Structure | SHD ↓ | F1 ↑ | SHD ↓ | F1 ↑ | SHD ↓ | F1 ↑ | SHD ↓ | F1 ↑ | SHD ↓ | F1 ↑ |
| Tree (LeakyReLU) | **0.67** (1.49) | **0.96** (0.08) | 5.83 (2.04) | 0.63 (0.09) | 6.00 (3.00) | 0.65 (0.08) | 7.50 (1.50) | 0.00 (0.00) | 11.83 (0.37) | 0.00 (0.00) |
| V-structure (LeakyReLU) | **0.67** (1.10) | **0.97** (0.05) | 7.67 (4.08) | 0.61 (0.14) | 5.50 (2.50) | 0.72 (0.08) | 8.00 (0.00) | 0.17 (0.17) | 17.33 (2.87) | 0.00 (0.00) |
| Tree (Tanh) | **1.00** (1.67) | **0.95** (0.09) | 5.50 (1.52) | 0.63 (0.06) | 4.50 (1.50) | 0.70 (0.03) | 7.50 (1.50) | 0.00 (0.00) | 16.50 (4.92) | 0.00 (0.00) |
| V-structure (Tanh) | **1.17** (1.33) | **0.95** (0.06) | 4.33 (2.58) | 0.79 (0.08) | 4.50 (1.50) | 0.76 (0.03) | 9.50 (1.50) | 0.36 (0.064) | 18.50 (3.25) | 0.00 (0.00) |

Note: SHD = Structural Hamming Distance (lower is better ↓). F1 scores range from 0 to 1 (higher is better ↑). Standard deviations are reported in parentheses.

The top layer typically captures global features, such as digit identity, while the middle layer learns variations within the same digit. The lowest layer focuses on local features that have minimal impact on the overall digit structure. Figure 3b visualizes a subset of the learned causal graph, highlighting these patterns, which were generated by intervening on different nodes in the sub-graph. The top node denotes the concept of a digit three. The lower nodes are visualized upon intervention. The full causal graph, shown in Appendix B.2, has 62 latent variables demonstrating the scalability of our method. Appendix B.2 also discusses visualization and intervention details.

Causal representations can contribute to better generalization and transfer learning due to the transfer of causal relations (Schölkopf et al., 2021). To demonstrate the effectiveness of our learned representations in domain transfer, we evaluate them on the CMNIST dataset (Arjovsky et al., 2019) and CelebA dataset (Liu et al., 2015). We describe the problem setting and results for CMNIST here, while CelebA is discussed in Appendix B.2. In this dataset, the training set consists of digits 0 and 1, colored either red or green. The color acts as a cause for the digit, with $P(\text{digit} = 0|\text{color} = \text{red}) = 0.9$ and $P(\text{digit} = 0|\text{color} = \text{green}) = 0.1$ in the training set. These probabilities are reversed in the test set. We further evaluate on an additional test set where all digits are colored blue. Samples from these sets are shown in Figure 3c. Although the correlation between color and digit varies across datasets, the causal relationship between the digit's image features and its label remains unchanged.

To predict the digit labels, we first learn a latent causal structure from the dataset. We then train a logistic regression classifier on these latent representations, applying L1 regularization to encourage sparsity in the model weights. This regularization promotes the use of features within the Markov blanket of the label. We evaluate the model on the test set, and the results are presented in Table 2.

We compare our method with standard Autoencoders, Causal VAEs (Yang et al., 2021), and Graph VAEs (He et al., 2018). For each baseline, a logistic regression classifier is trained on the learned la-

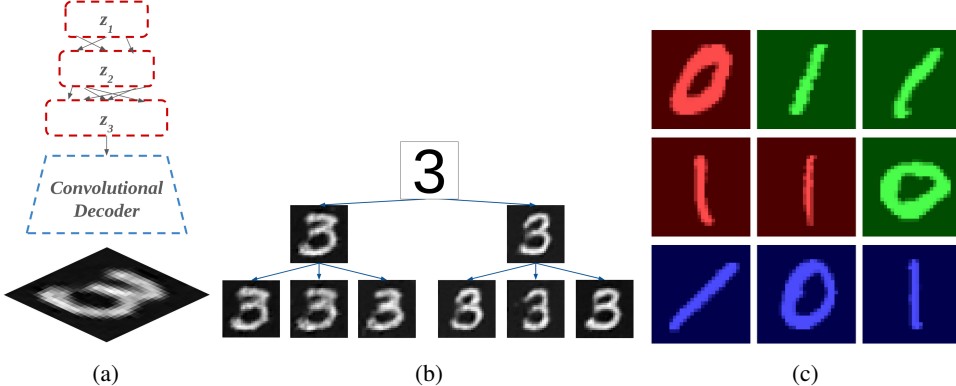

Figure 3: Figures for the Image experiments. (a) Latent causal graph for digit images (b) Visualization of subgraph of the learnt latent causal graph on MNIST (c) Samples from the CMNIST dataset illustrating digit-label associations under different conditions. Top row: training set samples with default color-label mapping. Middle row: test set samples with reversed color-label mapping. Bottom row: test set samples with a consistent blue color irrespective of labels.

Table 2: Test Accuracy on the CMNIST dataset. Standard deviations are reported in parentheses.

|  | Ours | Autoencoder | Graph VAE | Causal VAE |
|---|---|---|---|---|
| Reverse | 0.979 (0.004) | 0.854 (0.037) | 0.665 (0.231) | 0.916 (0.075) |
| Blue | 0.753 (0.106) | 0.649 (0.195) | 0.766 (0.174) | 0.653 (0.183) |

tent representations. All methods are evaluated over three random trials, with the mean and standard deviation reported to assess performance consistency.

Table 2 compares performance on the two test sets: 'Reverse,' where the color-digit relationship is reversed in the test set, and 'Blue,' where all digits in the test set are blue. While our representations demonstrate better transferability under distribution shifts compared to the evaluated baselines, we acknowledge that task-specific methods not focused on identifiable representation learning might have the potential to achieve higher accuracy.

## 7 CONCLUSION

In this work, we introduce a differentiable causal discovery method for recovering the structure of latent hierarchical causal graphs under rather mild conditions. Our approach significantly outperforms existing baselines and is scalable to high-dimensional datasets such as images. Additionally, we establish novel identifiability results without imposing restrictive assumptions on the structural equation models. Notably, our result that provides graphical information based on the rank of the Jacobian matrix may inspire future work in this area.

Despite relaxing several key assumptions, certain conditions, such as the two pure children requirement, remain necessary, as seen in other latent hierarchical methods (Huang et al., 2022; Kong et al., 2023). Furthermore, our method does not yet account for structures where measured variables have children. Future work could extend our identifiability results to more general structures, similar to Dong et al. (2023). We also speculate that Condition 3 may not be necessary, and future research could focus on relaxing this condition further.

## ACKNOWLEDGMENTS

The authors would like to thank the anonymous reviewers for valuable comments and suggestions. We would also like to thank Yan Li, Xiaofei Wang, Minghao Fu, and Fan Feng for their helpful comments on the paper. We would like to acknowledge the support from NSF DMS 2428058, NSF Award No. 2229881, AI Institute for Societal Decision Making (AI-SDM), the National Institutes of Health (NIH) under Contract R01HL159805, and grants from Quris AI, Florin Court Capital, and MBZUAI-WIS Joint Program.

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

# A PROOFS

## A.1 PROOF OF PROPOSITION 1

**Proposition 1.** *The probability of a distribution P generated by a structural model with respect to G violating Generalized Faithfulness is zero.*

*Proof.* We begin our proof with the following lemma.

**Lemma 5.** *Let $A \in \mathbb{R}^{m \times p}$ and $B \in \mathbb{R}^{p \times n}$ be random matrices drawn from a continuous distribution with $m, n \geq p$, whose entries are drawn from continuous distributions. Let $C = AB$ be their product. Then,*
$$\mathbb{P}(rank(C) < p) = 0$$
*with respect to the Lebesgue measure on $\mathbb{R}^{m \times n}$.*

*Proof.* The lebesgue measure of non-full rank matrices is zero. Therefore $\mathbb{P}(\text{rank}(A) = p) = 1$ and $\mathbb{P}(\text{rank}(B) = p) = 1$.

Since $C = AB$, $rank(C) \leq min(rank(A), rank(B)) = p$. For $rank(C) < p$, the vectors $Ab_{:,i}$ would have to be linearly dependent. This adds hard constraints on the matrices which has lebesgue measure zero.

$\square$

Using the proof of Theorem 1, we know $J_f(\boldsymbol{x}) = J_h(g(\boldsymbol{x})) \cdot J_g(\boldsymbol{x})$ where $J_h(g(\boldsymbol{x})) \in \mathbb{R}^{|\mathbb{Y}| \times |\mathbb{Z}|}, J_g \in \mathbb{R}^{|\mathbb{Z}| \times |\mathbb{X}|}$. Condition 3 ensures the Jacobian matrices are continuous. By Lemma 5, the probability of $rank(J_f(\boldsymbol{x})) < |\mathbb{Z}|$ is zero for all $\boldsymbol{x}$. $\square$

## A.2 PROOF OF THEOREM 1

**Theorem 1.** *Let $\mathcal{G}$ be a hierarchical latent causal model satisfying Condition 1. For any two sets of measured variables $\mathbb{X}$ and $\mathbb{Y}$ in $\mathcal{G}$, let $f(\boldsymbol{x}) = \mathbb{E}[\boldsymbol{y}|\boldsymbol{x}]$. Under the faithfulness and differentiability conditions, for any $r < |\mathbb{X}|, |\mathbb{Y}|$, the rank of the Jacobian matrix $\mathbf{J}_f = \frac{\partial f}{\partial \boldsymbol{x}} = r$ if and only if the size of the smallest set of latent variables that d-separates $\mathbb{X}$ from $\mathbb{Y}$ is $r$. Formally,*
$$rank(\mathbf{J}_f) = \min_{\mathbb{Z}} |\mathbb{Z}| \quad \text{such that} \quad \mathbb{X} \perp\!\!\!\perp_{\mathcal{G}} \mathbb{Y}|\mathbb{Z} \tag{11}$$
*where $\mathbb{Z}$ is a subset of latent variables in $\mathcal{G}$, and $\perp\!\!\!\perp_{\mathcal{G}}$ denotes d-separation in the graph $\mathcal{G}$.*

*Proof.* Let $\mathbb{X}$ and $\mathbb{Y}$ be two sets of measured variables in $\mathcal{G}$. By the structure of the hierarchical latent causal model and Condition 1, there are no direct edges between measured variables. Therefore, any d-separation between $\mathbb{X}$ and $\mathbb{Y}$ must be mediated through a set of latent variables.

Let $\mathbb{Z}$ be the minimal set of latent variables that d-separates $\mathbb{X}$ from $\mathbb{Y}$, i.e.,
$$\mathbb{X} \perp\!\!\!\perp_{\mathcal{G}} \mathbb{Y} \mid \mathbb{Z}.$$
This implies that the conditional distribution satisfies
$$p(\boldsymbol{y}|\boldsymbol{x}) = \int p(\boldsymbol{y}|\boldsymbol{z})p(\boldsymbol{z}|\boldsymbol{x})d\boldsymbol{z}.$$
Taking expectations, we obtain
$$\mathbb{E}[\boldsymbol{y}|\boldsymbol{x}] = \int \boldsymbol{y}\, p(\boldsymbol{y}|\boldsymbol{x})d\boldsymbol{y} \tag{12}$$
$$= \int \boldsymbol{y} \left( \int p(\boldsymbol{y}|\boldsymbol{z})p(\boldsymbol{z}|\boldsymbol{x})d\boldsymbol{z} \right) d\boldsymbol{y} \tag{13}$$
$$= \int \left( \int \boldsymbol{y}\, p(\boldsymbol{y}|\boldsymbol{z})d\boldsymbol{y} \right) p(\boldsymbol{z}|\boldsymbol{x})d\boldsymbol{z} \tag{14}$$
$$= \int \mathbb{E}[\boldsymbol{y}|\boldsymbol{z}]\, p(\boldsymbol{z}|\boldsymbol{x})d\boldsymbol{z}. \tag{15}$$

By Condition 3, there exists a differentiable function $g : \mathbb{R}^{|\mathbb{X}|} \to \mathbb{R}^{|\mathbb{Z}|}$ such that

$$p(\boldsymbol{z}|\boldsymbol{x}) = p(\boldsymbol{z}|g(\boldsymbol{x})).$$

Substituting this into the expectation, we have

$$\mathbb{E}[\boldsymbol{y}|\boldsymbol{x}] = \int \mathbb{E}[\boldsymbol{y}|\boldsymbol{z}] \, p(\boldsymbol{z}|g(\boldsymbol{x}))d\boldsymbol{z} = h(g(\boldsymbol{x})),$$

where we define the function $h : \mathbb{R}^{|\mathbb{Z}|} \to \mathbb{R}^{|\mathbb{Y}|}$ by

$$h(\boldsymbol{w}) = \int \mathbb{E}[\boldsymbol{y}|\boldsymbol{z}] \, p(\boldsymbol{z}|\boldsymbol{w})d\boldsymbol{z}.$$

Applying the chain rule to the composition of functions, the Jacobian of $\mathbb{E}[\boldsymbol{y}|\boldsymbol{x}]$ with respect to $\boldsymbol{x}$ is

$$J_f(\boldsymbol{x}) = J_h(g(\boldsymbol{x})) \cdot J_g(\boldsymbol{x}),$$

where $J_h(g(\boldsymbol{x}))$ is an $|\mathbb{Y}| \times |\mathbb{Z}|$ matrix and $J_g(\boldsymbol{x})$ is an $|\mathbb{Z}| \times |\mathbb{X}|$ matrix.

Using the rank inequality for matrix multiplication, we have

$$\mathrm{rank}(J_f(\boldsymbol{x})) \leq \min\left(\mathrm{rank}(J_h(g(\boldsymbol{x}))), \mathrm{rank}(J_g(\boldsymbol{x}))\right).$$

Therefore,

$$\mathrm{rank}(J_f(\boldsymbol{x})) \leq \min(|\mathbb{Z}|, |\mathbb{X}|, |\mathbb{Y}|)$$

Proposition 1 implies that the Jacobian achieves its maximal possible rank almost everywhere. Thus, using Condition 2,

$$\mathrm{rank}(J_f(\boldsymbol{x})) = \min(|\mathbb{Z}|, |\mathbb{X}|, |\mathbb{Y}|)$$

Therefore, $J_f = \frac{\partial f}{\partial \boldsymbol{x}} = r < |\mathbb{X}|, |\mathbb{Y}|$ if and only if $|\mathbb{Z}| = r$. This completes the proof.

**Linear Case:** Note that linear latent hierarchical models (Huang et al., 2022) are a special case of this theorem. If the causal relationship between $y$ and $z$ is linear, we have: $\mathbb{E}[\boldsymbol{y}|\boldsymbol{x}] = \mathbb{E}[\boldsymbol{y}|\mathbb{E}[\boldsymbol{z}|\boldsymbol{x}]]$. Therefore, $\mathbb{E}[\boldsymbol{z}|\boldsymbol{x}]]$ being a continuous differentiable function of $\boldsymbol{x}$ suffices and we do not require Condition 3 (ii).

$\square$

## A.3 Proof of Theorem 2

**Theorem 2.** *Let $\mathcal{G}$ be a hierarchical latent causal model satisfying Condition 1. Let $\mathbb{Z}_X, \mathbb{Z}_Y \subseteq \mathbb{Z}^i$ be two disjoint subsets of latent variables in some layer $\mathbb{Z}^i$ in $\mathcal{G}$, i.e., $\mathbb{Z}_X \cap \mathbb{Z}_Y = \emptyset$. Let $\mathbb{X}$ be the set of measured variables that are d-separated by $\mathbb{Z}_X$ from all other measured variables in $\mathcal{G}$ and let $\mathbb{Y}$ be the set of measured variables that are d-separated by $\mathbb{Z}_Y$ from all other measured variables in $\mathcal{G}$ such that $\mathbb{X} \cap \mathbb{Y} = \emptyset$. Then,*

$$r(\mathbb{Z}_X, \mathbb{Z}_Y) = r(\mathbb{X}, \mathbb{Y})$$

*Proof.* Since $\mathbb{Z}_X$ and $\mathbb{Z}_Y$ d-separate $\mathbb{X}$ and $\mathbb{Y}$ from all other variables, we know that $\mathbb{X}$ are the measured pure descendants of $\mathbb{Z}_X$, and $\mathbb{Y}$ are the measured pure descendants of $\mathbb{Z}_Y$. Further, $\mathbb{X} \cap \mathbb{Y} = \emptyset$. Therefore, if $\mathbb{Z}$ d-separates $\mathbb{Z}_X$ and $\mathbb{Z}_Y$, $\mathbb{Z}$ d-separates $\mathbb{X}$ and $\mathbb{Y}$. Moreover, if $\mathbb{Z}$ d-separates $\mathbb{X}$ and $\mathbb{Y}$ then given our structure, it must d-separate $\mathbb{Z}_X$ and $\mathbb{Z}_Y$.

This completes the proof. $\square$

## A.4 Proof of Lemma 1

**Lemma 1.** *Let $\mathcal{G}$ be a graph satisfying Conditions 1. A set of measured variables $\mathbb{S}$ are pure children of the same parent if and only if for any subset $\mathbb{T} \subseteq \mathbb{S}$, $r(\mathbb{T}, \mathbb{X} \setminus \mathbb{T}) = 1$.*

*Proof.* ($\Rightarrow$) Suppose the measured variables in $\mathbb{S}$ are pure children of the same parent node $p$. For any subset $\mathbb{T} \subseteq \mathbb{S}$, $\mathbb{T}$ and $\mathbb{X} \setminus \mathbb{T}$ are d-separated by node $p$. Therefore, we have $r(\mathbb{T}, \mathbb{X} \setminus \mathbb{T}) = 1$.

($\Leftarrow$) We prove the contrapositive. Suppose the union of parents of measured variables in $\mathbb{S}$ contains more than one node. Then there exist distinct parent nodes $p$ and $p'$ such that at least one of their respective children is in $\mathbb{S}$. We can choose $\mathbb{T}$ such that both $\mathbb{T}$ and $\mathbb{X} \setminus \mathbb{T}$ contains at least one pure child of $p$ and one pure child of $p'$. This choice is possible due to Condition 1, which states that all latent variables have at least two pure children.

For this choice of $\mathbb{T}$, we have $r(\mathbb{T}, \mathbb{X} \setminus \mathbb{T}) \geq 2$, as both $p$ and $p'$ are needed to d-separate the two sets. This contradicts the condition that $r(\mathbb{T}, \mathbb{S} \setminus \mathbb{T}) = 1$ for all $\mathbb{T} \subseteq \mathbb{S}$. □

### A.5    PROOF OF LEMMA 2

**Lemma 2.** *Let $\mathcal{G}$ be a hierarchical latent causal model satisfying Conditions 1. Under the generalized faithfulness condition, for any measured variable $c \in \mathbb{X}$ and any set of latent variables $\mathbb{P} \subseteq \mathbb{Z}^1$, $c$ is a child of exactly the variables in $\mathbb{P}$ if and only if the following conditions hold:*

1. *For each $\mathbb{S} \subseteq \mathbb{X}$ such that $|\mathbb{S} \cap Ch(z_i)| = 1$ for each $z_i \in \mathbb{P}$, where $Ch(z_i)$ denotes the set of pure children of $z_i$:*

$$r(\mathbb{S}, \mathbb{X} \setminus (\mathbb{S} \cup \{c\})) = r(\mathbb{S} \cup \{c\}, \mathbb{X} \setminus (\mathbb{S} \cup \{c\}))$$

2. *The equality in condition (1) does not hold for any proper subset of $\mathbb{P}$.*

*Proof.* We will prove both directions of the if and only if statement.

($\Rightarrow$) Suppose $c$ is a child of exactly the variables in $\mathbb{P}$.

Let $\mathbb{S} \subseteq \mathbb{X}$ be any set such that $|\mathbb{S} \cap Ch(z_i)| = 1$ for each $z_i \in \mathbb{P}$, and let $\mathbb{T} = \mathbb{X} \setminus (\mathbb{S} \cup \{c\})$.

Let $\mathbb{U} \subseteq \mathbb{Z}$ be the set of latent variables which d-separates $\mathbb{S}$ and $\mathbb{T}$. By the structure of the graph, $\mathbb{P} \subseteq \mathbb{U}$. Since $\{c\}$ is child of $\mathbb{P}$, $\mathbb{P}$ d-separates $\{c\}$ from $\mathbb{T}$. Therefore, $\mathbb{U}$ also d-separates $\mathbb{S} \cup \{c\}$ from $\mathbb{T}$. Therefore, we have $r(\mathbb{S}, \mathbb{T}) = r(\mathbb{S} \cup \{c\}, \mathbb{T})$.

This satisfies condition (1). Condition (2) is satisfied because no proper subset of $\mathbb{P}$ contains all parents of $c$, so no proper subset of $\mathbb{P}$ can d-separate $\mathbb{S} \cup \{c\}$ from $\mathbb{T}$.

($\Leftarrow$) Now suppose conditions (1) and (2) hold. We will prove that $c$ is a child of exactly the variables in $\mathbb{P}$.

Consider $\mathbb{S}$ constructed with exact one child of each latent node in $\mathbb{P}$. Therefore, $r(\mathbb{S}, \mathbb{T}) = |\mathbb{P}|$. Let $\mathbb{U} \subseteq \mathbb{Z}$ be the set which d-separates $\mathbb{S} \cup \{c\}$ and $\mathbb{T}$. It follows that $\mathbb{P} \subseteq \mathbb{U}$.

The equality in condition (1) implies:

$$|\mathbb{U}| = r(\mathbb{S} \cup \{c\}, \mathbb{T}) = r(\mathbb{S}, \mathbb{T}) = |\mathbb{P}|$$

Therefore, $\mathbb{P} = \mathbb{U}$ and $\mathbb{P}$ d-separates $c$ from $\mathbb{T}$. This implies that the parents of $c$ must be a subset of $\mathbb{P}$, as any path from $c$ to $\mathbb{T}$ not going through $\mathbb{P}$ would violate the d-separation.

Now, condition (2) states that this equality doesn't hold for any proper subset of $\mathbb{P}$. This means that every variable in $\mathbb{P}$ is necessary for the d-separation. If any variable in $\mathbb{P}$ were not a parent of $c$, then we could remove it and still maintain the d-separation, contradicting condition (2).

Therefore, $c$ must be a child of exactly the variables in $\mathbb{P}$. □

### A.6    PROOF OF LEMMA 3

**Lemma 3.** *Let $\mathcal{G}$ be a graph satisfying Conditions 1. A measured variable $c$ has no parent if and only if $r(\{c\}, \mathbb{X} \setminus \{c\}) = 0$.*

*Proof.* We will prove both directions of the if and only if statement.

($\Rightarrow$) Suppose $c$ has no parent.

In this case, $c$ is independent of all other variables in the graph. Therefore, $r(\{c\}, \mathbb{X} \setminus \{c\}) = 0$.

($\Leftarrow$) Suppose $r(\{c\}, \mathbb{X} \setminus \{c\}) = 0$.

This means that no latent variables are needed to render $c$ independent of all other observed variables. In other words, $c$ is already independent of all other observed variables.

Now, suppose for the sake of contradiction that $c$ has a parent $z$. By Condition 1, $z$ must have at least two pure children, one of which could be $c$, and let's call the other one $x$. Then $c$ and $x$ would be dependent through their common parent $z$, contradicting the independence of $c$ from all other observed variables.

Therefore, $c$ cannot have any parent. $\qquad\square$

### A.7 PROOF OF THEOREM 3

**Theorem 3.** *Let $\mathcal{G} = (\mathcal{V}, \mathcal{E})$ be a hierarchical latent causal model satisfying Condition 1. Let $\boldsymbol{M}$ be the binary adjacency matrix representing the structure of $\mathcal{G}$. Let data $\mathbb{X}$ be generated according to the structural equation model defined in Equation 1. Given a function $r(\mathbb{S}, \mathbb{T})$ which outputs the minimum number of latent variables which d-separate any two sets measured sets $\mathbb{S}$ and $\mathbb{T}$, $\boldsymbol{M}$ is identifiable up to the permutation of the latent variables.*

*Proof.* We prove this theorem by induction on the number of layers in the hierarchical latent causal model.

**Base case:** Let $\mathcal{G} = (\mathcal{V}, \mathcal{E})$ be a hierarchical latent causal model with one latent layer, i.e., $\mathbb{Z} = \mathbb{Z}^1$.

We begin by identifying the pure children of each latent variable in $\mathbb{Z}^1$. By Lemma 1, for any set of measured variables $\mathbb{S} \subseteq \mathbb{X}$, if $r(\mathbb{T}, \mathbb{X} \setminus \mathbb{T}) = 1$ for all $\mathbb{T} \subseteq \mathbb{S}$, then all variables in $\mathbb{S}$ are pure children of the same parent. For all $|\mathbb{T}| = 1$, this trivially holds. Using Theorem 1, we can exhaustively check this condition for all subsets $|\mathbb{T}| > 1$ of $\mathbb{X}$ to identify all sets of pure children.

Next, we identify the parents of non-pure children using Lemma 2. For each observed variable $c \in \mathbb{X}$ that is not identified as a pure child in the previous step, we determine its parents by verifying the conditions stated in Lemma 2 for all possible subsets of $\mathbb{Z}^1$. For most cases, $|\mathbb{S} \cup \{c\}|, |\mathbb{X} \setminus (\mathbb{S} \cup \{c\})| > |\mathbb{P}|$ which allows us to use Theorem 1. For cases, where this does not hold, it implies $\mathbb{P} = \mathbb{Z}^1$. In this case, Lemma 2 can be applied for all other subsets of $\mathbb{Z}^1$ and if none of them satisfy the conditions, the set of parents has to be $\mathbb{Z}^1$.

Through these two steps, we fully identify the structure between $\mathbb{Z}^1$ and $\mathbb{X}$, thus recovering the binary adjacency matrix $\boldsymbol{M}$ for the one-layer model.

**Inductive step:** Assume the theorem holds for all models with $L - 1$ layers, where $L > 1$. We will prove it holds for models with $L$ layers.

Let $\mathcal{G} = (\mathcal{V}, \mathcal{E})$ be a hierarchical latent causal model with $L$ layers. We first identify the structure between $\mathbb{Z}^1$ and $\mathbb{X}$ using the same process as in the base case, applying Lemmas 1 and 2.

Let $\mathcal{G}' = (\mathcal{V}', \mathcal{E}')$ be the sub-graph of $\mathcal{G}$ obtained by removing all measured variables $\mathbb{X}$. We claim that $\mathcal{G}'$ satisfies Condition 1. Each latent variable in $\mathbb{Z}^2$ has at least two pure children in $\mathcal{G}'$, and these belong to $\mathbb{Z}^1$. Moreover, the equal path length condition is preserved since the path from any latent to any variable in $\mathbb{Z}^1$ is one less than the path to any variable in $\mathbb{X}$. Some variables $\mathbb{Z}^1$ may not have any parent. We can identify those using Lemma 3.

To determine the d-separation relations between variables in $\mathbb{Z}^1$, we utilize Theorem 2. For any two subsets $\mathbb{Z}_X, \mathbb{Z}_Y \subseteq \mathbb{Z}^1$, let $\mathbb{X}$ and $\mathbb{Y}$ be sets of their respective pure children. By Theorem 2, we have $r(\mathbb{Z}_X, \mathbb{Z}_Y) = r(\mathbb{X}, \mathbb{Y})$, allowing us to infer the d-separation relations within $\mathbb{Z}^1$.

Now, $\mathcal{G}'$ is a hierarchical latent causal model with $L - 1$ layers that satisfies the conditions of the theorem. By the induction hypothesis, we can identify the structure of $\mathcal{G}'$, recovering the corresponding part of the binary adjacency matrix $\boldsymbol{M}$.

Therefore, by the principle of mathematical induction, the theorem holds for hierarchical latent causal models with any number of layers.

$\square$

**Note:** The identifiability up to a permutation of the latent variables means up to the relabeling of the latent variables. For example, consider a scenario with three latent variables representing gender, hair color, and facial hair, where gender causally influences both hair color and facial hair. If we assign $Z_1$ = gender, $Z_2$ = hair color, and $Z_3$ = facial hair, the causal structure is $Z_2 \leftarrow Z_1 \rightarrow Z_3$. Alternatively, if we assign $Z_1$ = hair color, $Z_2$ = facial hair, and $Z_3$ = gender, the structure becomes $Z_1 \leftarrow Z_3 \rightarrow Z_2$. Despite these different representations, the underlying causal relationships remain semantically identical.

### A.8 PROOF OF LEMMA 4

**Lemma 4.** *Consider a DAG $\mathcal{G}$ with a binary adjacency matrix $\boldsymbol{M}$. $\mathcal{G}$ satisfies Condition 1 (i) if and only if:*

$$\|\boldsymbol{M}_{i,:} \odot \prod_{j \neq i}(1 - \boldsymbol{M}_{j,:})\|_1 \geq 2 \quad \forall i. \tag{16}$$

*Proof.* To prove this lemma, we need to demonstrate that the given condition on the adjacency matrix $\boldsymbol{M}$ holds if and only if each latent variable has at least two pure children, as stated in Condition 1.

Let's first analyze the term inside the $\ell_1$ norm:

$$\boldsymbol{M}_{i,:} \odot \prod_{j \neq i}(1 - \boldsymbol{M}_{j,:})$$

The $k$th element of this vector is given by:

$$M_{ik} \cdot \prod_{j \neq i}(1 - M_{jk})$$

This product equals 1 if and only if $M_{ik} = 1$ and $M_{jk} = 0$ for all $j \neq i$. In other words, this term is 1 if and only if the vertex $v_k$ is a child of $v_i$ and not a child of any other $v_j$ ($j \neq i$). Hence, this product identifies whether $v_k$ is a pure child of $v_i$.

The $\ell_1$ norm, $\|\cdot\|_1$, sums up all these terms, meaning it counts the number of pure children of $v_i$.

Now, according to Condition 1, each latent variable $v_i$ must have at least 2 pure children. Therefore, the condition:

$$\|\boldsymbol{M}_{i,:} \odot \prod_{j \neq i}(1 - \boldsymbol{M}_{j,:})\|_1 \geq 2 \quad \forall i$$

ensures that each $v_i$ has at least 2 pure children, satisfying Condition 1.

Conversely, if each latent variable $v_i$ has at least 2 pure children, the sum $\left\|\boldsymbol{M}_{i,:} \odot \prod_{j \neq i}(1 - \boldsymbol{M}_{j,:})\right\|_1$ must be at least 2 for each $i$, proving the equivalence.

Thus, the lemma is proven.

$\square$

## B EXPERIMENTAL DETAILS

In this section, we provide a detailed explanation of our experimental setup, model architecture, training procedure, hyperparameter settings, and evaluation metrics used in the paper.

### B.1 SYNTHETIC DATA

We follow the same procedure and hyperparameter values for all four graphs.

**Model architecture:** We use a VAE to learn our causal structure as described in Section 5. The model consists of several components. The VAE encoder is a two-hidden-layer fully connected neural network with 64 and 32 hidden neurons, followed by ReLU activations. The encoder outputs both the mean ($\mu$) and the log variance ($\log \sigma^2$) for the latent variables. For the decoder, each function in Equation 4 is modeled using a one-hidden-layer fully connected neural network with 32 hidden neurons. The masking matrices have shape $\left\lfloor \frac{|\mathbb{X}|}{2^{i+1}} \right\rfloor \times \left\lfloor \frac{|\mathbb{X}|}{2^i} \right\rfloor$ since Condition 1 allows a maximum of $\left\lfloor \frac{|\mathbb{X}|}{2^i} \right\rfloor$ latent variables in $\mathbb{Z}^i$. We use ReLU activation for all neural networks.

**Training Procedure:** We use mean squared error as the reconstruction loss. We calculate the $\mathcal{L}_{\text{ind}}(\epsilon) = \text{KL}(p(\epsilon) \| \prod_j \prod_i p(\varepsilon_{z_j^i}) \prod_j p(\varepsilon_{x_j}))$ using a method similar to MINE (Belghazi et al., 2018). We warm up the MINE model for 100 epochs before training. Our model is trained using the Adam optimizer with a learning rate of $1 \times 10^{-3}$ for 400 epochs. We use a batch size of 32. For Gumbel-Softmax, we set the temperature to 1.0 throughout training. The coefficient for $\mathcal{L}_{\text{ind}}(\epsilon)$ is set to 10. $\lambda_2$ is set to 1e-4 and $\lambda_3$ is set as $10^{-3+\frac{\text{epoch}}{100}}$. Since the training objective is non-convex, we may get suboptimal solutions (Ng et al., 2024b). Therefore, we run the model with 10 random initializations and select the one with the lowest loss.

**Baselines:** For Kong et al. (2023), we use the code shared with us by the authors. For Huang et al. (2022) and Xie et al. (2020), we use the publicly available implementation. Default hyperparameters were used for all methods. We attempted to use FOFC (Kummerfeld & Ramsey, 2016) as a baseline, however an error occurred for our data since it does not satisfy the conditions for their code to run.

**Evaluation Metrics:** We evaluate our model using the Structural Hamming Distance (SHD) and F1 score between the learned adjacency matrix and the ground truth. We train each run three times for different seed values and report the mean and standard deviation across the three runs of two graphs. Since the latent graph may only be recovered up to a permutation of the latent variables, we calculate the SHD over all possible $|\mathbb{Z}|!$ permutations of the estimated graph and select the lowest SHD. Since the evaluation time is $O(n!)$ in the number of latent variables, evaluating methods becomes intractable for very large graphs. The time was calculated in seconds. For the standard deviation of time, we report the mean of the standard deviation of each graph since time can vary a lot based on the size of the graph.

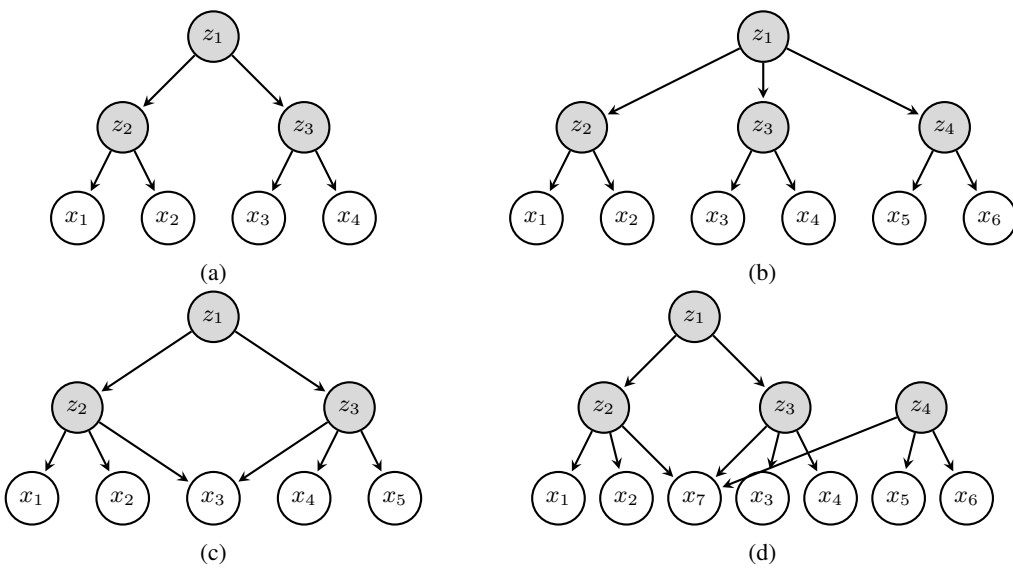

Figure 4: Ground truth causal graphs for Synthetic Experiments. (a) and (b) are trees (only one path between any two nodes). (c) and (d) allow v-structures (multiple paths between two nodes)

Table 3: Performance of latent hierarchical causal discovery methods on additional randomly generated graphs

| Ours | | KONG | | HUANG | | GIN | | DeCAMFounder | |
|---|---|---|---|---|---|---|---|---|---|
| SHD ↓ | F1 ↑ | SHD ↓ | F1 ↑ | SHD ↓ | F1 ↑ | SHD ↓ | F1 ↑ | SHD ↓ | F1 ↑ |
| **1.00** | **0.94** | 4.60 | 0.68 | 5.50 | 0.65 | 7.60 | 0.05 | 13.8 | 0.00 |
| (1.41) | (0.08) | (1.58) | (0.09) | (3.56) | (0.12) | (1.80) | (0.15) | (2.30) | (0.00) |

Note: SHD = Structural Hamming Distance (lower is better ↓). F1 scores range from 0 to 1 (higher is better ↑). Standard deviations are reported in parentheses.

**Additional Experiments:**   To further validate our methodology, we randomly sample 10 causal structures, applying our method alongside the baselines. We generate graphs with the number of variables ranging from 7 to 13, ensuring the structures satisfy Condition 1. Since the latent graph can only be recovered up to a permutation of the latent variables, we compute the Structural Hamming Distance (SHD) over all possible $|\mathbb{Z}|!$ permutations of the estimated graph and select the permutation with the lowest SHD. Due to the factorial complexity, $O(|\mathbb{Z}|!)$, of this evaluation in the number of latent variables, the computation becomes intractable for large graphs.

We report SHD and F1 scores in Table 3. We see significant improvement over baselines, in line with Table 1.

**Evolution of loss:**   In Figure 5, we plot the loss versus epochs for one of our training runs corresponding to the causal structure in Figure 4a. Metrics such as SHD and F1 are omitted, as they can only be computed once the mask values converge to either 0 or 1.

We see the structural regularization loss decreases over training and converges to zero, enforcing Condition 1. The ELBO loss initially decreases but then slightly increases as we enforce the structural constraint.

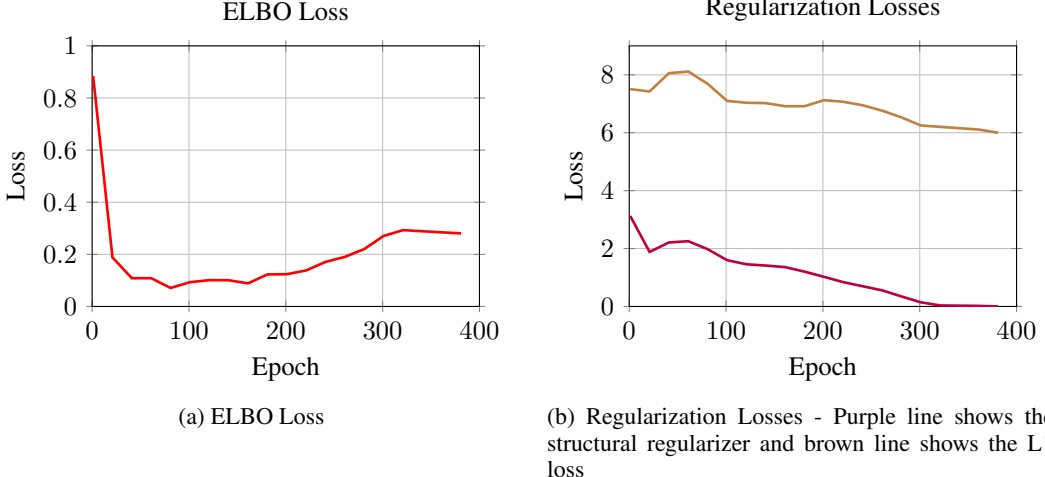

(a) ELBO Loss

(b) Regularization Losses - Purple line shows the structural regularizer and brown line shows the L1 loss

Figure 5: Evolution of different loss components during training

### B.2   IMAGE DATA

**Model Architecture:**   The proposed Hierarchical VAE model consists of a convolutional encoder, a hierarchical latent structure, and a transposed convolutional decoder. The convolutional encoder has two convolution layers (32 and 64 filters, 3x3 kernels, stride 2) followed by a fully connected layer. Each structural equation (Equation 4) is modeled using a neural network with three hidden layer. The first two layers are shared to reduce the number of parameters. The convolutional decoder

reconstructs images from the final latent layer. The decoder consists of a fully connected layer that maps the final latent representation of dimension 49 to a 1568-dimensional space. This output is then reshaped to a 32-channel 7x7 feature map. We then have two transposed convolutional layers with 32 and 16 filters respectively (both using 3x3 kernels, stride 2, padding 1, and output padding 1), and a final transposed convolutional layer that reconstructs the image. We initialize $M$ with three layers with 10, 20 and 49 nodes in each of the three layers. We use ReLU activation for all neural networks.

**Training:** We train the model on a subset of 10,000 images. The model was trained for 300 epochs. We use batch size of 256 and do not use MINE to enforce independence between the exogenous variables. We find it makes little difference to the final output. The temperature for gumbel softmax starts at 100 and exponentially decreases to 0.1 at 120 epochs and then stays constant. $\lambda_2$ is 0.03 and $\lambda_3$ is exponentially increased from $10^{-3}$ to 10 at 100 epochs and then stays constant. We used a Adam optimizer with learning rate 1e-3.

**Visualization:** Figure 6 displays the complete causal graph constructed from the MNIST dataset. Note that due to non-convexity, we could not achieve zero loss for the pure children constraint term. Hence, the learnt graph does not exactly satisfy Conditions 1. To interpret the semantics of each latent feature, we perform targeted interventions designed to isolate their individual effects. Specifically, for each latent variable, we set its ancestral nodes to values 5 standard deviations above or below their means, while keeping the remaining variables at their mean values. We then intervene on the current node by setting its value to the mean, effectively neutralizing its direct influence, and observe the resulting changes in the generated images. This procedure allows us to visualize and understand the specific contribution of each latent variable to the overall image structure.

By comparing the images before and after the intervention, we can discern the unique effects attributable to each latent feature. Table 5 presents these visualizations for each feature. For each feature, the first image shows the output when the top latent variable is set 5 standard deviations below the mean; the second image shows the result when, in addition, the target feature is intervened upon and set to the mean; the third image displays the output when the top latent variable is set 5 standard deviations above the mean; and the fourth image shows the result when the target feature is intervened upon and set to the mean under this condition.

For Figure 3b, we adopt a different methodology to visualize the influence of latent variables. Starting from the topmost layer of the causal graph, we traverse downward through each subsequent layer. At each node, we randomly assign its value to be either five standard deviations above or below its mean. This stochastic intervention allows us to observe the cumulative effects of these variations as they propagate through the graph.

**Discussion on Learned MNIST Graph:** As detailed above, Table 5 provides visualizations for all latent features. The contrast between the first and second images (or between the third and fourth images) illustrates the concept represented by each feature. We observe that the hierarchical structure of the learned latent variables effectively captures features at different levels of abstraction. Specifically, the top layer encodes global features (e.g., digit-level information), the middle layer encodes intermediate features, and the bottom layer captures local characteristics.

For instance, consider $z_8^0$: the difference between the first and second images reveals that this feature represents the digit 9. A positive value of $z_8^0$ correlates with the presence of the digit 9. The children of $z_8^0$ in the causal graph, $z_6^1$ and $z_{13}^1$, further decompose this concept into its components. The difference between their respective visualizations shows that $z_6^1$ represents the straight line in the lower half of the digit 9, while $z_{13}^1$ captures the circular component in the upper half, along with the overall thickness of the digit.

Continuing this decomposition, the children of $z_6^1$ and $z_{13}^1$, such as $z_4^2$, $z_{10}^2$, $z_{11}^2$, and $z_{16}^2$, represent finer-grained local features. This hierarchical organization aligns with the semantics of the images and supports the interpretability of the latent representations.

**CMNIST details:** For the colored MNIST dataset, we have around 12,000 training samples and 2,000 test samples. Since we do not aim to visualize the images, we downsample them to 14x14 and train our model. We train our model for 50 epochs with early stopping with patience 3. After

Table 4: Test AUC on the CelebA dataset

|        | Ours      | Graph VAE | Causal VAE |
|--------|-----------|-----------|------------|
| CelebA | **0.823** | 0.500     | 0.729      |

training the latent hierarchical model, we train a logistic regression classifier to predict the digit from the latent representations. The coefficient of the L1 regularization is 10. For all the baselines, we train the model for 50 epochs with early stopping with patience 3. For all models, we used a Adam optimizer with learning rate 1e-3.

**CelebA details:**  For the CelebA dataset, we consider the task of predicting whether a face image has blonde hair. Since gender and hair color are highly correlated, models often use gender to predict hair color. In this task, we reverse the correlation of gender and color and test the transferability of representations.

We use approximately 160,000 samples for training. For the test set, we evaluate the model exclusively on two groups: blonde males and non-blonde females. Since visualization of the images is not a focus of this work, we downsample all images to a resolution of 64×64.

The model is trained for 50 epochs, employing early stopping with a patience of 3 epochs. After training the latent hierarchical model, we fit a logistic regression classifier on the latent representations to predict the target attribute. The coefficient for the L1 regularization in the logistic regression is set to 10. For all baseline models, we adopt the same training setup of 50 epochs with early stopping (patience = 3). We use the Adam optimizer with a learning rate of $10^{-3}$ for all models.

**CelebA Results:**  The results are available in Table 4. Our model achieves a test AUC of **0.8228**, outperforming both Graph VAE (AUC 0.500) and Causal VAE (AUC 0.7289). The results highlight the transferability of our representations.

## C  DISCUSSION ON CONDITION 1

In Figure 7, we see two examples of causal graphs which violate Condition 1. Figure 7a violates the pure children condition since each latent does not have two pure children. Figure 7b violates the Condition 1(ii) since $d(z_1, x_1) = 3 \neq 2 = d(z_1, x_3)$. While Condition 1(ii) may not hold in all cases, it is a reasonable assumption to make for image data. Several prior works Vahdat & Kautz (2020); Kong et al. (2024) have effectively modeled images using a multi-level latent hierarchical structure.

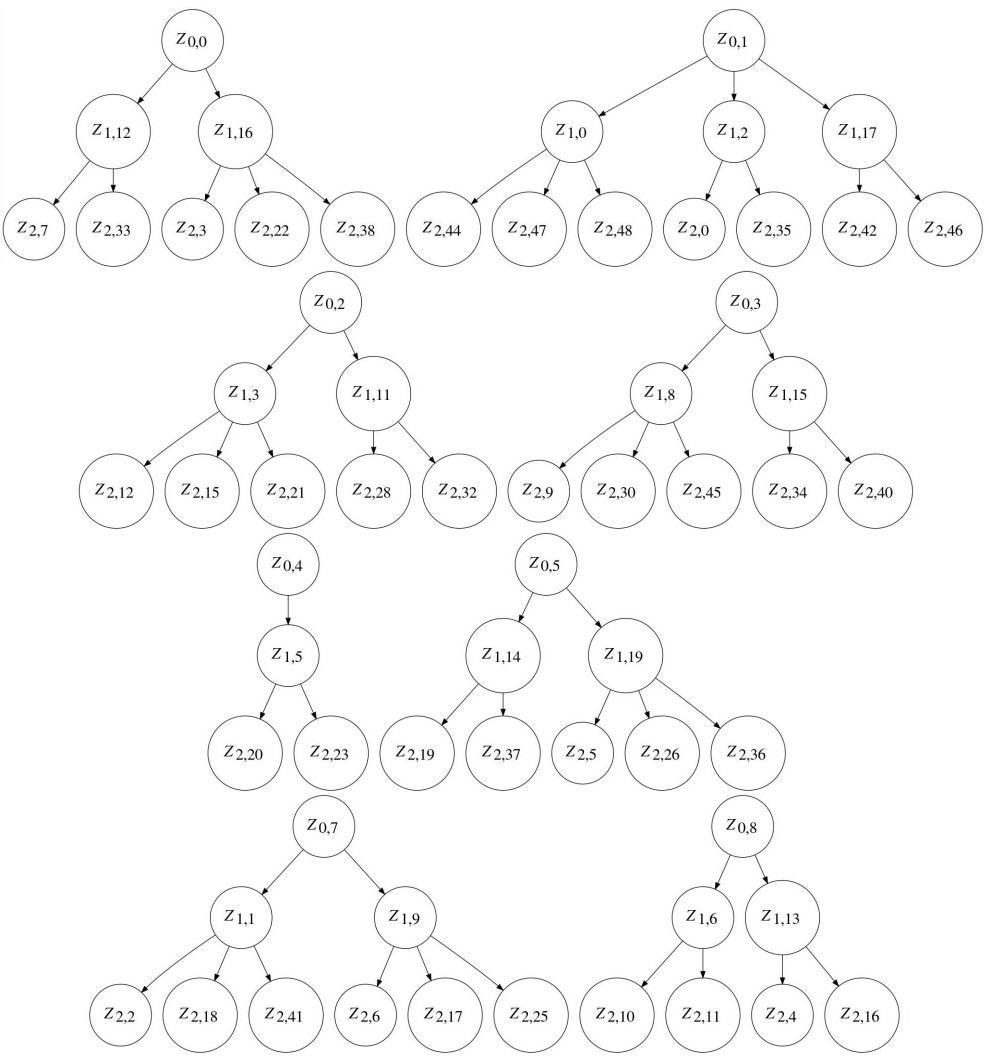

Figure 6: Latent causal graph for the MNIST Dataset. $z_{i,j}$ denotes the $j^{\text{th}}$ latent variable in $\mathbb{Z}^i$.

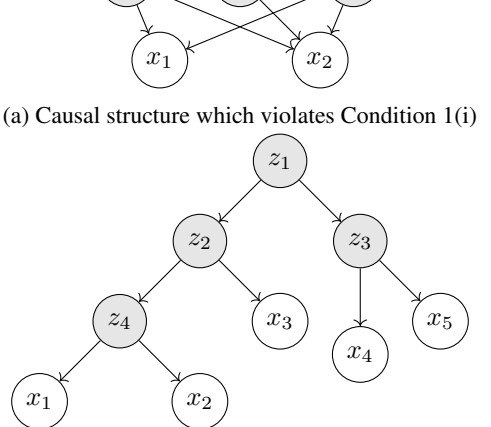

(a) Causal structure which violates Condition 1(i)

(b) Causal structure which violates Condition 1(ii)

Figure 7: Two examples of causal structures which violate Condition 1

Table 5: Visualization of MNIST layer dimensions. For each feature, the first image shows the output when the top latent variable is set 5 standard deviations below the mean; the second image shows the result when, in addition, the target feature is intervened upon and set to the mean; the third image displays the output when the top latent variable is set 5 standard deviations above the mean; and the fourth image shows the result when the target feature is intervened upon and set to the mean under this condition.

| Dim | Image | Dim | Image | Dim | Image |
|-----|-------|-----|-------|-----|-------|
| $z_0^0$ |  | $z_1^0$ |  | $z_2^0$ |  |
| $z_3^0$ |  | $z_4^0$ |  | $z_5^0$ |  |
| $z_7^0$ |  | $z_8^0$ |  | $z_0^1$ |  |
| $z_1^1$ |  | $z_2^1$ |  | $z_3^1$ |  |
| $z_5^1$ |  | $z_6^1$ |  | $z_8^1$ |  |
| $z_9^1$ |  | $z_{11}^1$ |  | $z_{12}^1$ |  |
| $z_{13}^1$ |  | $z_{14}^1$ |  | $z_{15}^1$ |  |
| $z_{16}^1$ |  | $z_{17}^1$ |  | $z_{19}^1$ |  |
| $z_0^2$ |  | $z_2^2$ |  | $z_3^2$ |  |
| $z_4^2$ |  | $z_5^2$ |  | $z_6^2$ |  |
| $z_7^2$ |  | $z_8^2$ |  | $z_9^2$ |  |
| $z_{10}^2$ |  | $z_{11}^2$ |  | $z_{12}^2$ |  |
| $z_{15}^2$ |  | $z_{16}^2$ |  | $z_{17}^2$ |  |
| $z_{18}^2$ |  | $z_{19}^2$ |  | $z_{20}^2$ |  |
| $z_{21}^2$ |  | $z_{22}^2$ |  | $z_{23}^2$ |  |
| $z_{24}^2$ |  | $z_{25}^2$ |  | $z_{26}^2$ |  |
| $z_{28}^2$ |  | $z_{30}^2$ |  | $z_{32}^2$ |  |
| $z_{33}^2$ |  | $z_{34}^2$ |  | $z_{35}^2$ |  |
| $z_{36}^2$ |  | $z_{37}^2$ |  | $z_{38}^2$ |  |
| $z_{40}^2$ |  | $z_{41}^2$ |  | $z_{42}^2$ |  |
| $z_{44}^2$ |  | $z_{45}^2$ |  | $z_{46}^2$ |  |
| $z_{47}^2$ |  | $z_{48}^2$ |  | | |

