# OpenReview forum: "Differentiable Causal Discovery for Latent Hierarchical Causal Models"
_ICLR.cc/2025/Conference — ICLR 2025 Poster_

### Official Review · Reviewer_ehtm · 2024-10-25

**Soundness:** 3
**Presentation:** 3
**Contribution:** 3
**Rating:** 8
**Confidence:** 4

**Summary:**

This paper introduces a novel differentiable causal discovery method for latent hierarchical causal models (LHCMs) and derives identifiability conditions of LHCMs in non-linear cases with relaxed assumptions (i.e., no requirement of invertible functions). In the experimental evaluation, the authors show promising results outperforming existing methods on synthetic and image data.

**Strengths:**

- paper is written clearly while keeping a formal discussion of assumptions and theorems
- the authors derive and prove their identifiability conditions. The proofs look correct after careful checking.
- a novel differentiable DAG learner for LHCMs is introduced, allowing differentiable structure learners to be applied in latent variable settings
- the experimental section introduces an interesting experiment on image data that demonstrates that the proposed algorithm can be seamlessly integrated into the autoencoder framework, thus allowing learning of LHCMs on complex and unstructured data such as images

**Weaknesses:**

**Section 2**
- the authors discuss "differentiable causal discovery" in the related work. However, most (if not all) works referenced here do not perform causal discovery. This has been shown by several works, e.g., [1], [2]

**Section 4**
While the theorems and proofs in Sec. 4 are correct, it is unclear to me whether the identifiable model still allows for a causal interpretation if variable permutations are allowed (Theorem 3). It would be good to clarify which permutations are allowed and why the permutations do not change the causal structure (and thus $d$-separation statements). To illustrate what I mean, consider a LHCM (where observed $X$ are dropped for the sake of simplicity) $Z_2 \leftarrow Z_1 \rightarrow Z_3$. If (any) permutation is allowed, Theorem 3 would also allow for $Z_1 \leftarrow Z_3 \rightarrow Z_2$. However, this model entails different $d$-separation statements and thus has different causal semantics. Hence the causal model would not be identifiable.

**Section 5**
It is unclear to me how the acyclicity and overall model structure from Condition (1) (ii) is ensured/reflected in the objective (if at all reflected) (Eq. 10). Based on this, it is not easy to see why the proposed method is not just a structure learner, but a causal discovery method. Could the authors please provide more details on how this is achieved?

**Section 6**
- Tab. 1: Why do the baselines perform so badly? Is there any specific explanation for that?
- Synthetic experiment: How were the ground truth structures chosen? By hand or randomly? If by hand, could the authors explain why and why these?
- image experiments: There is the work on causalVAEs [3], why did you not choose this as a baseline? Since it is more related to the overall problem setup of this work than standard VAEs, this baseline would make much sense.

# References
[1] Reisach et al. Beware of the simulated dag! causal discovery benchmarks may be easy to game. NeurIPS 2021.

[2] Seng et al. Learning Large DAGs is Harder Than You Think. ICLR 2024.

[3] Yang et al. CausalVAE: Structured Causal Disentanglement in Variational Autoencoder. 2020.

**Questions:**

see weaknesses

# Additonal Notes
Note that I decided on a score of 6 as there is no option 7. If the authors address the points in the weaknesses section accordingly, I'm inclined to raise my score.

---

> ### Author Response · Authors · 2024-11-21
> **Rebuttal by Authors**
>
> We thank the reviewer for their constructive feedback and suggestions to improve our paper. We appreciate that the reviewer has recognized the clarity of our writing, the correctness of the theoretical results, and the novelty of introducing a differentiable DAG learner for latent hierarchical causal models (LHCMs). Below, we address the concerns and provide clarifications.
>
> **Section 2: Do papers on differentiable causal discovery referenced in the paper actually perform causal discovery?**
>
> We agree that some of the referenced works in the related work section do not fully perform causal discovery, as highlighted by [1] and [2]. We have updated the manuscript to explicitly discuss these limitations. However, we note that the critiques in [1] are not universally applicable. Improved evaluation protocols, such as using sampling with non-equal noise variances, can mitigate these issues and provide robust results for proposed causal discovery approaches [4]. These practices have been incorporated into our experimental evaluations.
>
> **Section 4: Identifiability upto permutation**
>
> Latent variables are inherently hidden, and their labels can only be identified up to a permutation. For instance, the structures `Z2 <- Z1 -> Z3` and `Z1 <- Z3 -> Z2` represent the same causal relationships, even though the meanings of `Z1`, `Z2`, and `Z3` differ. This is because the d-separation properties, which define the causal semantics, remain unchanged under such permutations.
>
> **Section 5: Enforcing structural constraints**
>
> Condition 1(i) is incorporated through Eqs. (6) and (8). These conditions are reflected in the final term of Eq. (10), which equals zero if and only if each latent variable has two pure children, ensuring that the structural constraints are satisfied.
>
> Condition 1(ii) is reflected in the block structure of the adjacency matrix `M`, which enforces acyclicity and consistency with the hierarchical structure.
>
> **Q: Causal Discovery vs. Structure Learning:**
> Enforcing structural constraints alone does not guarantee recovery of the true causal graph, as multiple graphs can satisfy these constraints. Our method addresses this by jointly optimizing for likelihood and sparsity while satisfying the constraints, enabling the discovery of the true causal graph.
>
> **Section 6**
> -**Tab. 1: Why do the baselines perform so poorly?**
>
> The baselines perform poorly because they rely on assumptions of linearity or deterministic relationships, which are not satisfied in our data. This highlights the effectiveness of our method in more general nonlinear settings.
>
> -**Synthetic Experiment: How were the ground truth structures chosen?**
>
> The ground truth structures were chosen randomly.
>
> -**Image Experiments: Why not compare with CausalVAE?**
>
> Thank you for pointing this out. CausalVAE [3] requires additional information in the form of concept labels, which our setting does not provide. To address this, we have included a comparison with a modified version of CausalVAE that does not use concept labels in the updated manuscript.
>
> **References**
>
> [1] Reisach, C., et al. "Beware of the simulated DAG! Causal discovery benchmarks may be easy to game." *NeurIPS,* 2021.
> [2] Seng, A., et al. "Learning Large DAGs is Harder Than You Think." *ICLR,* 2024.
> [3] Yang, G., et al. "CausalVAE: Structured Causal Disentanglement in Variational Autoencoder." *NeurIPS,* 2020.
> [4] Ng, I., Huang, B., & Zhang, K. "Structure learning with continuous optimization: A sober look and beyond." *Causal Learning and Reasoning, PMLR,* 2024.
>
> Please let us know if you have further concerns, and please consider raising the score if we have cleared existing concerns – thank you so much!

---

> > ### Comment · Reviewer_ehtm · 2024-11-22
> > **Response to Rebuttal**
> >
> > Thank you for the detailed response to my review.
> >
> > Most of my concerns have been resolved.
> >
> > The only concern/question still unresolved for me is whether the learned graphs can be interpreted as causal graphs.  Let me explain why:  As you said in the rebuttal, the meaning of the latent variables (and so the labels) can differ, leading to permutation invariance. I agree that Eq. 6 & 8 encode the structural constraints imposed on the graph to be learned. However, given the permutation invariance, a causal interpretation is not possible without further assumptions. To stick to the example above, `Z2 <- Z1 -> Z3` and `Z1 <- Z3 -> Z2` have causally different meanings. In the first case, Z1 causes Z2, and in the second case, Z3 causes Z2. I would appreciate it if the authors could try to clarify how the permutation invariance of latents aligns with the causal interpretation of the learned graph.
> >
> > Nevertheless, the paper's quality increased in the revised version, and new baselines have been added. Thus, I increased my score to 8.

---

> > > ### Author Response · Authors · 2024-11-22
> > >
> > > Thank you for your thoughtful response and for increasing your score. We greatly appreciate your engagement with our work and your constructive feedback.
> > >
> > > Regarding the interpretation of permutation invariance and its alignment with causal interpretation, we would like to clarify further using an example. Consider a latent graph (we drop observed variables for simplicity) where **hair color ← gender → facial hair**. Since all variables are latent, their labels can be arbitrarily assigned. For instance:
> > >
> > > 1. If `Z1 = gender`, `Z2 = hair color`, and `Z3 = facial hair`, the causal graph is `Z2 ← Z1 → Z3`.
> > > 2. Alternatively, if `Z1 = hair color`, `Z2 = facial hair`, and `Z3 = gender`, the causal graph is `Z1 ← Z3 → Z2`.
> > >
> > > While the labeling of the latent variables may differ, the **semantics of the causal relationships and the underlying structure of the graph remain consistent**.
> > >
> > > We hope this explanation clarifies the alignment between permutation invariance and the causal interpretation of the learned graph. Thank you again for your detailed feedback and support!

---

> > > > ### Comment · Reviewer_ehtm · 2024-11-23
> > > >
> > > > Thank you for the example, now the invariance makes sense to me!
> > > > To improve the clarity of your works, I think it would be good to add such an example in the Appendix.

---

> ### Author Response · Authors · 2024-11-26
> **Official Comment by Authors**
>
> Thank you for your reply and continued discussion. We are happy we could clarify the causal interpretation of permutation in variance. In order to increase clarity, we will add an example in the Appendix as per your suggestion.
>
> Thanks again for your time and engagement! We highly appreciate this opportunity to exchange opinions and discuss with you.

---

### Official Review · Reviewer_sGn5 · 2024-11-02

**Soundness:** 2
**Presentation:** 3
**Contribution:** 2
**Rating:** 5
**Confidence:** 4

**Summary:**

Differentiable causal discovery has been a key focus of the causality community in the past years. Despite the advance of representation learning and deep learning, differentiable hierarchical causal discovery with latent variables has been a challenging subfield with at least empirical limited results and limited impact despite the need and call for these methods from practical applications.

The paper proposes a new method and investigates some of the conditions for identifiability.

While the paper has some very interesting and promising components, I overall can not recommend it for acceptance in its current form.

**Strengths:**

I really like that the evaluation is not just done with respect to a causal metric but wrt to "a regression classifier trained on the learned representation". If the causality field would move towards the standard evaluation practices of deep learning progress would be faster and this paper is one of the few which actually does perform this evaluation!
However, when reading the paper in more detail e.g. Table 1 is then again evaluated wrt to discovery metrics only table 2 is evaluated with a learned classifier and arguably table 2 provides only a very limited setting and very limited evaluation. Especially given that these are deep learning approaches, the performance should not even reported in a table but as plots where the x-axis is training time and the y-axis performance. This would account for complexity and cost of training and really allow for a fair comparison of the approaches.

While it is argued that causal representations lead to better generalizations and transfers this is so far actually not shown in the literature. DomainBed and or [1] clearly state the need for better evaluation and clearer demonstrations of the benefits beyond deriving identifiability results. I am thus really encouraging the authors to significantly extend the ablations and plot train vs performance curves and the performance of the classifier at different stages of training in a larger scale setting and across significantly more datasets.

[1] Saengkyongam, Sorawit, et al. "Identifying representations for intervention extrapolation." arXiv preprint arXiv:2310.04295 (2023).

**Weaknesses:**

The key claimed advantage for better identifiability results comes from the fact that instead it is assumed that "not yet account for structures where measured variables have children"

There is some exchangeability of these assumptions and in that sense I agree that the current assumption is a more practical one but it is not a novel one or a clear contribution until a clear relation between the assumptions is shown.

The evaluation is really lacking wrt to datasets and shown clear benefits across different settings. As mentioned I think the authors already take a very valuable step for the community by not only evaluating wrt to discovery metrics (see strengths) but adopting the established evaluation frameworks in deep learning of training a classifier on top of a learned representation. However that evaluation is unfortunately severely limited.

**Questions:**

It seems that the baselines are chosen from one lab only i.e. Xie et al, Kong et al and Huang et al which are used to sell the method are all from one lab.

Given the number of baselines available for the task that seems a bit strange. Can you please clarify?

---

> ### Author Response · Authors · 2024-11-21
> **Rebuttal by Authors**
>
> We thank the reviewer for their constructive feedback and insightful comments. This feedback helps strengthen our paper. Below, we address each of the concerns raised and provide clarifications.
>
> **W1: Assumptions and Contributions**
>
> **Q:** There is some exchangeability of these assumptions and in that sense I agree that the current assumption is a more practical one but it is not a novel one or a clear contribution until a clear relation between the assumptions is shown.
>
> **A:** Thank you for your comments. We would like to emphasize the novelty of our work and its connections to previous studies below:
>
> Most existing causal discovery methods assume the absence of causal relations between latent variables. Among the few methods that permit such relations, to the best of our knowledge, they are either limited to **linear** models [1][2] or **deterministic mappings** [3] (e.g., where  $z = f(x)$  and $f$  is invertible). For instance, the method proposed by Kong et al. [3] requires that latent variables can be expressed as an invertible function of observed variables, which excludes even simple relationships like  $X = \sin(Z) + \epsilon$ . These constraints significantly limit their applicability in real-world scenarios.
>
> In contrast, our work introduces a more general and practical framework for modeling latent hierarchical graphs, relaxing the aforementioned restrictive assumptions. Specifically:
> - **General Latent Relations:** Our framework supports non-linear and non-deterministic causal relationships, broadening the scope of latent structures that can be identified. This is a significant departure from existing approaches that are constrained by linearity or invertibility.
> - **Jacobian Rank Indicator for d-Separation:** We establish a novel Jacobian rank indicator to characterize d-separation in latent hierarchical graphs. Using this indicator, we provide a rigorous proof of identifiability under our relaxed assumptions. To the best of our knowledge, this contribution is original and represents a non-trivial theoretical advancement in causal discovery.
> - **Practical and Theoretical Implications:** By enabling the modeling of general latent hierarchical graphs, our work overcomes practical limitations in existing methods, paving the way for applications to more complex real-world datasets. Additionally, the Jacobian rank approach has potential for extension to other graph classes, opening new directions for future research.
>
> We believe that relaxing restrictive assumptions and establishing identifiability for a broader class of latent graphs represent a significant and novel contribution to the field. These advancements address fundamental gaps in the existing literature and offer practical value for applications beyond current methods.
>
> **Q:** The key claimed advantage for better identifiability results comes from the fact that instead it is assumed that "not yet account for structures where measured variables have children.
>
> **A:** First we would like to clarify that our key contribution to the identifiability results is the development of a **novel Jacobian-rank indicator** for determining the number of d-separating latent variables in the non-linear case. This allows us to handle nonlinear, non-deterministic, and non-invertible latent causal relations (see Theorem 1 in Section 4).
>
> Furthermore, these results can indeed be extended to account for structures where measured variables have children, as suggested by the work of Dong et al. [1], which builds upon Huang et al. [2] for linear models.
>
> Dong et al. demonstrate that, with an appropriate indicator for the number of variables d-separating any two observed variables, it is possible to recover the causal graph even when measured variables have children, under weak structural assumptions. We acknowledge this as a potential extension of our work, which we briefly discussed in Section 7 of the manuscript.

---

> ### Author Response · Authors · 2024-11-21
> **Rebuttal by Authors Continued**
>
> **W2: Experimental Evaluations**
>
> **Real Data Evaluation**
>
> We thank the reviewer for appreciating the diversity of our evaluation methods. To further strengthen the evaluation, we have performed additional experiments:
> - Added experiments on the **CelebA** dataset. Results are in Appendix B.2 Table 4.
> - Incorporated two additional baselines: **CausalVAE** [6] and **GraphVAE** [7], which explicitly model latent causal structures. Results are in Section 6 Table 2.
>
> **Q1: Baselines**
>
> Our method addresses the identifiability of **nonlinear latent hierarchical models**, a setting with very limited comparable baselines. Most existing methods do not allow relations between latent variables or require strict assumptions. **We have included the baselines which model latent hierarchical structures. These baselines are closest to our setting.**
>
> To provide a more comprehensive comparison, we have included **DeCAMFounder** [4] as a baseline in the updated manuscript. DeCAMFounder founder does not model latent hierarchical structures but learns causal graphs even in the presence of latent confounding. However, as it does not allow relations between latent variables, it performs poorly in our setting.
>
> **FOFC** [5] is another causal discovery method which aims to discover causal structures with latent variables. FOFC does not allow relations between latent variables and requires each latent variable to have at least three pure children (in contrast we require only two). We attempted to compare with FOFC, but its strict requirement for three pure children per latent variable is not satisfied by our dataset.
>
> Our approach shows substantial improvements in Structural Hamming Distance (SHD) and F1 scores over all baselines, confirming its effectiveness.
>
> Please refer to the general response for more details on additional experiments.
>
> **Q2: Additional Visualizations and Plots**
>
> We appreciate the feedback on incorporating additional visualizations. To address this concern:
> - We included a **plot of performance vs. computational time** for each method in Figure 2. However, we would like to clarify that most baselines compared to are not deep learning approaches since there do not exist any for latent hierarchical causal discovery.
> - We included loss vs epoch plots in Appendix B.1 Figure 5 as suggested by the reviewer.

---

> > ### Author Response · Authors · 2024-11-24
> > **Rebuttal by Authors Continued**
> >
> > **On evaluation and focus of the paper**
> >
> > We emphasize the main contributions of our paper:
> > 1. **Novel Identifiability Results**: As far as we know, this is the first work to prove the identifiability of latent hierarchical causal models without assuming linear or deterministic relations.
> > 2. **Practical Methodology**: We propose a differentiable causal discovery method for latent hierarchical models, addressing scalability and error propagation issues of discrete search methods.
> >
> > Our evaluations in this paper aim to validate these contributions. Since our paper's main goal is causal discovery, we extensively evaluate our proposed method on metrics which are common and standard across causal discovery literature. Since most real-world datasets lack ground truth causal graphs, causal discovery methods are typically evaluated on synthetic data. In our updated manuscript, we extended synthetic experiments to include additional nonlinear activation functions and new baselines. **Our method significantly outperforms all baselines. Moreover, we are considerably faster compared to Kong et al [3] which is the only other non-linear hierarchical baseline.**
> >
> > In order to demonstrate scalability, we also learn causal graphs for Image data. However, since the ground truth graph is not available, we evaluate our graph using indirect methods. Using MNIST data, we demonstrate that our model learns interpretable representations across layers and that these representations are useful for transfer learning compared to latent variable methods that do not learn such structures. While we include additional causal representation learning baselines, we do not claim state-of-the-art performance on transfer learning tasks. We have clarified this in the updated manuscript.
> >
> > As mentioned by the reviewer, our step toward evaluating causal methods using additional metrics beyond discovery metrics is valuable. We plan to thoroughly investigate these metrics in future work. However, in this paper, our primary focus is on relaxing key assumptions in the causal discovery literature, providing theoretical contributions, and proposing a scalable differentiable causal discovery approach.
> >
> >
> > **References**
> >
> > [1] Dong, X., et al. "On the Parameter Identifiability of Partially Observed Linear Causal Models." *arXiv preprint arXiv:2407.16975* (2024).
> > [2]Huang, Biwei, et al. "Latent hierarchical causal structure discovery with rank constraints." Advances in neural information processing systems 35 (2022): 5549-5561.
> > [3] Kong, L., et al. "Identification of nonlinear latent hierarchical models." *Advances in Neural Information Processing Systems* 36 (2023): 2010-2032.
> > [4] Agrawal, R., et al. "The DeCAMFounder: nonlinear causal discovery in the presence of hidden variables." *Journal of the Royal Statistical Society Series B: Statistical Methodology* 85.5 (2023): 1639-1658.
> > [5] Kummerfeld, E., and Ramsey, J. "Causal clustering for 1-factor measurement models." *Proceedings of the 22nd ACM SIGKDD International Conference on Knowledge Discovery and Data Mining.* 2016.
> > [6] Yang, M., et al. "CausalVAE: Disentangled representation learning via neural structural causal models." *Proceedings of the IEEE/CVF Conference on Computer Vision and Pattern Recognition.* 2021.
> > [6] He, J., et al. "Variational autoencoders with jointly optimized latent dependency structure." *International Conference on Learning Representations.* 2019.
> >
> > Please let us know if you have further concerns, and please consider raising the score if we have cleared existing concerns – thank you so much!

---

> > > ### Comment · Reviewer_sGn5 · 2024-11-25
> > > **Thanks for rebuttal.**
> > >
> > > Key point from the authors "However, in this paper, our primary focus is on relaxing key assumptions in the causal discovery literature, providing theoretical contributions, and proposing a scalable differentiable causal discovery approach." There are so many identifiability results that it is really hard to obtain a good overview how the approaches connect or relate. Due to the unlimited number of combinations of assumptions, it is a sheer endless list of possible papers where it is hard to evaluate real progress.
> > > Given that the assumptions are not strictly holding in practice and are at best crude approximations of reality (in the end it is embedded in a deep learning model) it is crucial to not only evaluate wrt discovery metrics which have inherent problems but to actually train on downstream tasks in addition. Does the identified causal graph actually help in something we are interested in? In table 2 that is actually done and that is really great and so I update my score but overall I am not convinced even after reading the reviews of the other authors.

---

> > > > ### Author Response · Authors · 2024-11-26
> > > > **Thanks for response.**
> > > >
> > > > Thank you for your thoughtful response and for increasing your score. We greatly appreciate your engagement with our work and your constructive feedback. We particularly appreciate the positive recognition of our empirical work.
> > > >
> > > > We would like to address the few lingering concerns of the reviewer.
> > > >
> > > > > Due to the unlimited number of combinations of assumptions, it is a sheer endless list of possible papers where it is hard to evaluate real progress.
> > > >
> > > > Thanks for the thoughtful comments; however, we respectfully disagree with this premise. Exploring assumptions for identifiability is not only essential for ensuring model robustness but also crucial for guiding model design. Moreover, while there could theoretically be an unlimited combination of assumptions when adding new ones, the focus of our paper is on **removing and relaxing assumptions** (such as linearity or invertibility). Since existing work relies on a finite set of assumptions, this concern does not apply to our approach.
> > > >
> > > > Moreover, we consider a widely studied class of models in this paper. Latent hierarchical models (or special cases of such models like trees, or 1-factor measurement models) have been extensively studied in prior work [1][2][3][4][5][6], but they were unable to establish identifiability in the general non-linear case. We generalize the setting of these papers and several of these works are special cases of our framework.
> > > >
> > > > > Given that the assumptions are not strictly holding in practice and are at best crude approximations of reality (in the end it is embedded in a deep learning model). Does the identified causal graph actually help in something we are interested in?
> > > >
> > > > Thanks for asking this question and allowing us to clarify applications of such models. These assumptions often do hold in interesting problems. We provide several examples where these assumptions are valid and useful:
> > > >
> > > > - **Gene Regulatory Networks (GRNs):** Gene expression data is observed, but transcriptional regulatory networks are latent. Latent hierarchical models help identify hidden regulators or shared biological function [7].
> > > > - **Image Data:** Generative models for image data are hypothesized to be compositional and hierarchical, with latent abstract concepts [8][9].
> > > > - **Complex Social Systems:** Hierarchical latent structures play a crucial role in understanding complex systems in political science and epidemiology [10]. For example, in epidemiology, clinical and microbiological findings are observed but disease states and population-level etiological agents are latent [11].
> > > >
> > > > For image data, we demonstrate the utility of our approach in terms of interpretability and transferability, as shown in Section 6.2 (Table 2) and Appendix B.2 (Table 4). Beyond image data, we believe our methodology would assist domain experts in fields such as genomics and sociology, enabling the discovery of latent causal graphs even when the data does not satisfy strict assumptions.
> > > >
> > > > Finally, we are uncertain about the reviewer’s statement that "in the end it is embedded in a deep learning model." While deep learning is used to model the generative process, the causal relations are explicitly parameterized using a masking matrix. During training, the parameters of this matrix converge to 0 or 1, explicitly revealing the causal relationships between the latent variables.
> > > >
> > > > We hope these clarifications address your concerns and highlight the broader applicability and robustness of our work.

---

> ### Author Response · Authors · 2024-11-26
> **Thanks for response Continued**
>
> ***References***
>
> [1]Anandkumar, Animashree, et al. "Learning linear bayesian networks with latent variables." International Conference on Machine Learning. PMLR, 2013.
>
> [2]Kummerfeld, Erich, and Joseph Ramsey. "Causal clustering for 1-factor measurement models." Proceedings of the 22nd ACM SIGKDD international conference on knowledge discovery and data mining. 2016.
>
> [3]Huang, Furong, et al. "Guaranteed scalable learning of latent tree models." Uncertainty in Artificial Intelligence. PMLR, 2020.
>
> [4]Adams, Jeffrey, Niels Hansen, and Kun Zhang. "Identification of partially observed linear causal models: Graphical conditions for the non-gaussian and heterogeneous cases." Advances in Neural Information Processing Systems 34 (2021): 22822-22833.
>
> [5]Huang, Biwei, et al. "Latent hierarchical causal structure discovery with rank constraints." Advances in neural information processing systems 35 (2022): 5549-5561.
>
> [6]Kong, Lingjing, et al. "Identification of nonlinear latent hierarchical models." Advances in Neural Information Processing Systems 36 (2023): 2010-2032.
>
> [7] Gitter, A., et al. "Unsupervised learning of transcriptional regulatory networks via latent tree graphical models." arXiv preprint arXiv:1609.06335 (2016).
>
> [8] Higgins, I., et al. "SCAN: Learning hierarchical compositional visual concepts." arXiv preprint arXiv:1707.03389 (2017).
>
> [9] Liu, N., et al. "Unsupervised compositional concepts discovery with text-to-image generative models." Proceedings of the IEEE/CVF International Conference on Computer Vision (2023).
>
> [10] Weinstein, E. N., & Blei, D. M. "Hierarchical Causal Models." arXiv preprint arXiv:2401.05330 (2024).
>
> [11] O'Brien, K. L., et al. "Causes of severe pneumonia requiring hospital admission in children without HIV infection from Africa and Asia: The PERCH multi-country case-control study." The Lancet 394.10200 (2019): 757-779.
>
> Please let us know if you have further concerns. We highly appreciate this opportunity to exchange opinions with you and learn from your perspective. Please kindly let us know your thoughts, and thank you again for your time and engagement!

---

> ### Author Response · Authors · 2024-12-01
> **Looking forward for Futher Discussion**
>
> We sincerely thank you for engaging with our rebuttal and participating in the discussion. We appreciate your time and valuable feedback. With the discussion period ending soon, we hope our response addresses your lingering concerns. We understand your busy schedule, but would greatly appreciate it if you could consider our updates when discussing with the AC and other reviewers.
>
> Thank you again for your thoughtful and constructive input!

---

### Official Review · Reviewer_bSAD · 2024-11-04

**Soundness:** 4
**Presentation:** 3
**Contribution:** 3
**Rating:** 8
**Confidence:** 3

**Summary:**

This paper shows that are particular class of causal graphs with  hierarchical latent variables are identifiable by leveraging properties of the Jacobian of the conditional exception function between subsets of observed variables. They then present an efficient algorithm for inferring the hierarchical graph. They present strong empirical results on both synthetic & image based problems.

**Strengths:**

- I thought this was interesting, original work. The class of graphs that they study is obviously limited but seems practical & the rank condition is intuitive.
- The paper is very well written - both the theory and methods section do a good job of explaining the intuition for why the method works
- The empirical results are strong on the datasets that they tested.

**Weaknesses:**

* The coloured MNIST results appear very strong (though this is not my area), but not contextualized in the domain generalization literature. I would have at least expected you to report the published numbers from recent work from that setting. Autoencoders & Beta-VAE is not the right baselines?
* I would have liked a more detailed discussion of the learned MNIST graph. I am not sure what to make of figure 4 or table 3 in the appendix? Do those latents make sense? Is there a natural hierarchical structure that we would expect?

**Questions:**

See weakness above.

---

> ### Author Response · Authors · 2024-11-21
> **Rebuttal by Authors**
>
> We thank the reviewer for their feedback and strong support of our work. We appreciate that the reviewer has recognized the originality and strong theoretical, methodological, and empirical contributions. Below, we address the concerns and provide clarifications.
>
> **Q1: Baselines for Colored MNIST Results**
>
> Thank you for highlighting this point. In the revised manuscript, we have included two additional baselines—**CausalVAE** [1] and **GraphVAE** [2]—that explicitly model the latent causal structure. These baselines provide a more meaningful comparison for our approach.
>
> Additionally, we clarified that our aim is not to achieve state-of-the-art performance on the Colored MNIST task but to evaluate the **transferability** of our learned representations in comparison to other representation learning methods.
>
> **Q2: Discussion of Learned MNIST Graph**
>
> We have added a detailed discussion of the learned MNIST graph in **Appendix B.2**. This includes an analysis of **Figure 4** and **Table 3**, clarifying how the latent variables align with an interpretable hierarchical structure.
>
> **References**
>
> [1] Yang, Mengyue, et al. "CausalVAE: Disentangled representation learning via neural structural causal models." *Proceedings of the IEEE/CVF Conference on Computer Vision and Pattern Recognition.* 2021.
>
> [2] He, J., et al. "Variational autoencoders with jointly optimized latent dependency structure." *International Conference on Learning Representations.* 2019.
>
> We hope these updates address the reviewer’s concerns. Please let us know if there are further points requiring clarification.

---

> > ### Author Response · Authors · 2024-12-01
> > **Gentle Reminder**
> >
> > We sincerely appreciate your time and valuable feedback. With the discussion period ending soon, we hope our responses address your concerns. We understand your busy schedule, but would greatly appreciate it if you could consider our updates when discussing with the AC and other reviewers.
> >
> > Thank you again for your thoughtful and constructive input!

---

### Official Review · Reviewer_wGhK · 2024-11-08

**Soundness:** 2
**Presentation:** 3
**Contribution:** 2
**Rating:** 6
**Confidence:** 3

**Summary:**

The main theoretical contribution of the paper is showing identifiability of nonlinear latent hierarchical causal models. Building on this theory, the authors propose a practical differentiable latent causal discovery approach. Experiments are performed on synthetic data as well as the coloured MNIST dataset to demonstrate efficacy of the approach.

**Strengths:**

1. The paper is, to the best of my knowledge, the first to provide identifiability results for nonlinear latent hierarchical causal models. The proof technique seems correct to me, though I did not check it thoroughly (for example, the appendix).

2. Estimating equation 9 using Donsker-Varadhan representation is novel.

**Weaknesses:**

1. **Experimental limitations**:

    a. **Synthetic experiments**: Instead of experimenting on just 4 structures given in figure 3, I would encourage authors to randomly generate DAGs and run experiments on these structures. For the synthetic experiments, the analysis would be stronger if the authors also try nonlinear activations for eq 1, instead of piecewise linear activation such as LeakyRELU.

   b. **Real experiments**: The baselines for the experiments on CMNIST are VAE and $\beta$-VAE -- both of which do not learn a structure over latent variables -- when  better baselines exist [1-3]. Applications to real world data is also limited, and even in the colored MNIST setting, only 2 digits seem to be used.

2. **Missing/weak motivation**: It is also unclear why such models are useful in the real world: motivation for why one needs such models would make the paper more strong. In the introduction, causal discovery is motivated but the there is no true causal structure for the CMNIST data. Given this, what is the purpose for obtaining a hierarchical structure as in Fig 2b? For what tasks, is such a hierarchical representation useful?

3. L447 - 453 mentions interventions but key details are missing regarding interventional data generation (single node or multi node interventions, soft vs hard intervention, and intervention values).

4. **Related work**: The task of causal discovery over latent variable hierarchical models is closely related to causal representation learning but this has not been discussed and  works in the space have not been cited [1, 2].

---

[1] Brehmer, Johann, et al. "Weakly supervised causal representation learning." Advances in Neural Information Processing Systems 35 (2022): 38319-38331.

[2] Subramanian, J. et al. Learning latent structural causal models. arXiv preprint arXiv:2210.13583 (2022)

[3] He, J. et al. Variational autoencoders with jointly optimized latent dependency structure. In International Conference on Learning Representations, 2019.

**Questions:**

1. What is the implication of condition 3?

2. There is a typo in equation 8, the number of small norms and large norms do not match.

3. From eq 6, we see that $|| M_{i, :} \odot \pi (1 - M_{j, :})||_1 \geq 2$.

However in the subject to constraint in eq 8, $||M_{i,:}||_1$ times the above entity is enforced to be $\geq 2$. This is a bit unclear -- can the authors clarify?

4. Caption for figure 2c is unclear.

PS: Score has been increased post rebuttal.

---

> ### Author Response · Authors · 2024-11-21
> **Rebuttal by Authors**
>
> We thank the reviewer for their constructive feedback and insightful comments. We appreciate that the reviewer has recognized the novelty and significance of our theoretical results. Below, we address each of the concerns raised and provide clarifications.
>
> **W1: Experimental Limitations**
> We have updated the experimental section of our manuscript in response to the concerns regarding the experiments. Please refer to the general response for details. In summary, we add experiments for each of the points the reviewer raised.
> - Added experiments using **tanh** as a non-linear activation function, complementing the results using piecewise linear **leakyReLU**. Results are in Section 6 Table 1.
> - Randomly generated a diverse set of **DAGs** and conducted experiments on these structures, providing robust and generalizable insights. Results are in Appendix B.1 Table 3.
> - Integrated **GraphVAE** [9] and **CausalVAE** [10] as additional baselines, showcasing comparative performance on latent causal structure learning. Results are in Section 6 Table 2.
> - Expanded the scope of real-world experiments to include the **CelebA dataset** for further validation. Results are in Appendix B.2 Section 4.
>
>
> **W2: Missing/Weak Motivation**
>
> We have clarified the applications of latent hierarchical causal models in the introduction of the revised manuscript. Below, we describe the importance of such models:
>
> Learning latent causal models can address critical challenges in **interpretability**, **distribution shifts**, and **scientific discovery** [1]. Latent hierarchical models are particularly relevant in domains such as:
>
> - **Gene Regulatory Networks (GRNs):** Gene expression data is observed, but transcriptional regulatory networks are latent [2].
> - **Image Data:** Generative models for image data are hypothesized to be compositional and hierarchical with latent abstract concepts [3][4].
> - **Complex social systems:** Hierarchical latent structures have been show to play a crucial role in understanding complex systems in political science and epidemiology. [5][6]
>
> Although latent hierarchical causal models have tangible real-world applications, the theoretical understanding of these models has been underexplored.  Existing works primarily demonstrate identifiability for **linear, discrete**, or **deterministic models**, making our contribution the first to establish identifiability for **general nonlinear hierarchical latent causal models**. Additionally, these methods use discrete search which is infeasible for high-dimensional data like images. We propose a differentiable approach and demonstrate that such latent representations can be learnt for high-dimensional data. While the true latent causal graph is unknown for real image data, our results showcase the **interpretability** and **transferability** of learned representations on datasets like MNIST and CelebA.
>
> **W3: Intervention Details**
>
> We have provided a detailed explanation of intervention data generation in Appendix B.2 under Visualization. To ensure clarity, we added a pointer to this section in the main text. Should we include a brief summary in the main text as well?
>
> **W4: References to Work in Causal Representation Learning**
>
> Thank you for pointing out relevant literature. We have updated the Related Work section to include and discuss [7][8][9][10].
>
> **Q1: Implication of Condition 3**
>
> Condition 3 ensures differentiability of the function ‘f’ and ‘g’ for theoretical results involving the rank of the Jacobian. This is a sufficient condition, though we do not believe it is necessary. Our proof (Appendix A.2) relies on the relationship `J_f = J_h(g(x)) J_g(x)`, where `p(z|x) = p(z|g(x))`. While we use leakyReLU in experiments, which is not differentiable, we believe future work can build on our results to relax this condition.
>
> **Q 2 & 3:Typo in Eq. (8) and Mismatch with Eq. (6)**
>
> We apologize for the typo in Eq. (8). This has been corrected in the revised manuscript, resolving the mismatch with Eq. (6).
>
> **Q4. Caption for Figure 2c**
>
> The caption for Figure 2c has been updated for clarity. Please let us know if it remains unclear.

---

> > ### Author Response · Authors · 2024-11-24
> > **Rebuttal by Authors Continued**
> >
> > **References**
> >
> > [1] Schölkopf, B., et al. "Toward causal representation learning." *Proceedings of the IEEE* 109.5 (2021): 612-634.
> > [2] Gitter, A., et al. "Unsupervised learning of transcriptional regulatory networks via latent tree graphical models." *arXiv preprint arXiv:1609.06335* (2016).
> > [3] Higgins, I., et al. "SCAN: Learning hierarchical compositional visual concepts." *arXiv preprint arXiv:1707.03389* (2017).
> > [4] Liu, N., et al. "Unsupervised compositional concepts discovery with text-to-image generative models." *Proceedings of the IEEE/CVF International Conference on Computer Vision* (2023).
> > [5] Weinstein, E. N., & Blei, D. M. "Hierarchical Causal Models." *arXiv preprint arXiv:2401.05330* (2024).
> > [6] O'Brien, K. L., et al. "Causes of severe pneumonia requiring hospital admission in children without HIV infection from Africa and Asia: The PERCH multi-country case-control study." *The Lancet* 394.10200 (2019): 757-779.
> > [7] Brehmer, J., et al. "Weakly supervised causal representation learning." *NeurIPS* (2022): 38319-38331.
> > [8] Subramanian, J., et al. "Learning latent structural causal models." *arXiv preprint arXiv:2210.13583* (2022).
> > [9] He, J., et al. "Variational autoencoders with jointly optimized latent dependency structure." *ICLR* (2019).
> > [10] Yang, Mengyue, et al. "Causalvae: Disentangled representation learning via neural structural causal models." Proceedings of the IEEE/CVF conference on computer vision and pattern recognition. 2021.
> >
> > Please let us know if you have further concerns, and please consider raising the score if we have cleared existing concerns – thank you so much!

---

> ### Author Response · Authors · 2024-11-26
> **Looking forward for Futher Discussion**
>
> We look forward to your thoughts on our response. Let us know if there is anything more we can do to address your comments!
>
> Thanks again for your time and constructive feedback!

---

> > ### Author Response · Authors · 2024-12-01
> > **Gentle Reminder**
> >
> > We sincerely appreciate your time and valuable feedback. With the discussion period ending soon, we hope our responses address your concerns. We understand your busy schedule, but would greatly appreciate it if you could consider our updates when revising your rating and discussing with the AC and other reviewers.
> >
> > Thank you again for your thoughtful and constructive input!

---

### Author Response · Authors · 2024-11-21
**General response**

We sincerely thank all reviewers for their constructive feedback and valuable suggestions. We appreciate the recognition of the originality, theoretical rigor, and clarity of our work. The reviewers’ feedback has helped us strengthen the paper. We have made changes to the manuscript, highlighted in blue for ease of reading. We address key changes and improvements in this response, with more details in the individual responses.

**Strengths Highlighted by Reviewers**

- **Novel Theoretical Contribution**: Our paper, to the best of our knowledge, is the first to establish identifiability results for nonlinear latent hierarchical causal models.  Our proofs are original and use novel techniques like jacobian rank indicator for d-separation. We also propose a practical, differentiable latent causal discovery algorithm, overcoming limitations of discrete search methods like error propagation and scalability. (Reviewer wGhK, Reviewer bSAD, Reviewer ehtm)
- **Empirical Evaluations**: Our experiments demonstrate that the proposed approach significantly outperforms baselines on causal discovery. Additionally, we validate our learned representations on real datasets, showcasing their interpretability and the utility of causal representations.  (Reviewer bSAD, Reviewer sGn5, Reviewer ehtm)
- **Well written**: Reviewers mention our paper is well-written and easy to follow with a formal discussion of assumptions and theorems. (Reviewer bSAD, Reviewer ehtm)



**Additional Experiments**

To address the reviewers' concerns, we have conducted several new experiments and added them to the updated manuscript:
- **Synthetic Data**:
   - Added experiments using `tanh` as the activation function alongside `LeakyReLU`.
   - Included  **DeCAMFounder** [1] as an additional baseline for a more comprehensive comparison.
   - The synthetic graphs used in the original paper were generated randomly. In the updated manuscript, we have extended the evaluation by including additional experiments on a wider range of randomly generated DAGs. The results are presented in Appendix B.1, Table 3.

Table 1: Performance of latent hierarchical causal discovery methods on various graphs

| Structure              | Ours (SHD ↓) | Ours (F1 ↑) | KONG (SHD ↓) | KONG (F1 ↑) | HUANG (SHD ↓) | HUANG (F1 ↑) | GIN (SHD ↓) | GIN (F1 ↑) | DeCAMFounder (SHD ↓) | DeCAMFounder (F1 ↑) |
|------------------------|--------------|-------------|--------------|-------------|---------------|--------------|-------------|------------|-----------------------|----------------------|
| Tree (LeakyReLU)       | **0.67**     | **0.96**    | 5.83         | 0.63        | 6.00          | 0.65         | 7.50        | 0.00       | 11.83                | 0.00                |
| V-structure (LeakyReLU)| **0.67**     | **0.97**    | 7.67         | 0.61        | 5.50          | 0.72         | 8.00        | 0.17       | 17.33                | 0.00                |
| Tree (Tanh)            | **1.00**     | **0.95**    | 5.50         | 0.63        | 4.50          | 0.70         | 7.50        | 0.00       | 16.50                | 0.00                |
| V-structure (Tanh)     | **1.17**     | **0.95**    | 4.33         | 0.79        | 4.50          | 0.76         | 9.50        | 0.36       | 18.50                | 0.00                |




2. **Real Data**:
   - Added comparisons with **CausalVAE** [2] and **GraphVAE** [3], explicitly modeling latent causal structures to strengthen our evaluation.



   - Expanded evaluation to include the **CelebA** dataset for broader empirical validation.

The detailed results for CMNIST and CelebA datasets are available in Section 6.2 Table 2 and Appendix B.2 Table 4 in the updated manuscript.

Table 2: Test Accuracy on the CMNIST dataset.
|                    | Ours   | Graph VAE | Causal VAE |
|--------------------|--------|-----------|------------|
| Reverse            | **0.979**  | 0.665     | 0.916      |
| Blue               | 0.753  | **0.766**     | 0.653      |

Table 4: Test AUC on the CelebA dataset.
|            | Ours   | Graph VAE | Causal VAE |
|---------|--------|-----------|------------|
| CelebA  | **0.8228** | 0.500     | 0.7289     |

---

> ### Author Response · Authors · 2024-11-21
> **General Response Continued**
>
> Note: The test set in CelebA is highly imbalanced (\(P(Y=1) = 0.97\)). Consequently, we use AUC as the evaluation metric instead of accuracy, which can be misleading in such scenarios. For instance, GraphVAE achieves an accuracy of 0.97 simply by predicting \(Y=1\) for every test point.
>
> We note that many other baselines, such as [4][5] (highlighted by Reviewer wGhK) and others [6][7], require auxiliary information, multiple domains, or interventional data, making them unsuitable for comparison in our setting.
>
> CausalVAE requires concept labels for identifiability as well. However, we were able to adapt their algorithm to run without concept labels. For GraphVAE, we could not find an official implementation from the authors, so we implemented the baseline ourselves. Despite this effort, we observed unstable training and generally poor performance compared to both CausalVAE and our proposed approach.
>
> **Motivation and Contribution of Our Work**
>
> Latent hierarchical causal models are found across various domains, including gene regulation, computer vision, political science, and epidemiology [8][9][10]. Despite their significance, there is a lack of theoretical understanding regarding the identifiability of these models in the general non-linear setting. Moreover, existing approaches for latent hierarchical causal discovery primarily rely on discrete search, which is computationally infeasible for high-dimensional data. Current causal representation learning methods often do not model hierarchical structures and require additional information, such as interventions or concept labels.
>
> Our contributions are twofold:  1)We prove identifiability results for general nonlinear latent hierarchical causal models.  2)We propose a differentiable approach that scales effectively to high-dimensional data.
>
> In the absence of ground truth causal structures for real datasets, causal discovery methods are typically evaluated using synthetic data. We follow this practice and demonstrate significant improvements over baselines on synthetic datasets. For real-world data, we indirectly validate the effectiveness of our approach by showcasing the interpretability and transferability of the learned representations.
>
> ***References***
>
> [1] Agrawal, R., et al. "The DeCAMFounder: nonlinear causal discovery in the presence of hidden variables." *Journal of the Royal Statistical Society Series B: Statistical Methodology* 85.5 (2023): 1639-1658.
>
> [2] Yang, M., et al. "CausalVAE: Disentangled representation learning via neural structural causal models." *Proceedings of the IEEE/CVF Conference on Computer Vision and Pattern Recognition.* 2021.
>
> [3] He, J., et al. "Variational autoencoders with jointly optimized latent dependency structure." *International Conference on Learning Representations.* 2019.
>
> [4] Brehmer, J., et al. "Weakly supervised causal representation learning." *Advances in Neural Information Processing Systems* 35 (2022): 38319-38331.
>
> [5] Subramanian, J., et al. "Learning latent structural causal models." *arXiv preprint arXiv:2210.13583* (2022).
>
> [6] Zhang, K., et al. "Causal representation learning from multiple distributions: A general setting." *arXiv preprint arXiv:2402.05052* (2024).
>
> [7] Hyvarinen, A., Sasaki, H., and Turner, R. "Nonlinear ICA using auxiliary variables and generalized contrastive learning." *The 22nd International Conference on Artificial Intelligence and Statistics.* PMLR, 2019.
>
> [8] Gitter, A., et al. "Unsupervised learning of transcriptional regulatory networks via latent tree graphical models." *arXiv preprint arXiv:1609.06335* (2016).
>
> [9] Higgins, I., et al. "SCAN: Learning hierarchical compositional visual concepts." *arXiv preprint arXiv:1707.03389* (2017).
>
> [10] Weinstein, E. N., and Blei, D. M. "Hierarchical Causal Models." *arXiv preprint arXiv:2401.05330* (2024).

---

### Meta-Review · Area_Chair_aWk8 · 2024-12-23

**Metareview:**

This paper attacks a longstanding problem in causal modeling: differentiability for hierarchical causal discovery. The authors propose a solution to this important problem which is motivated by theoretical results on identifiability of nonlinear latent hierarchical causal models.

__Strengths:__
1. The authors tackle a significant and important problem
2. The research is solid: from theory, to insights and then to a proposed solution.
3. The experimental evaluation (after the rebuttal) is convincing.

__Weaknesses:__
The main weakness (brought up by one reviewer) is with regards to training and evaluation on downstream tasks.

While more evidence of the method's utility could be provided for downstream tasks, this paper largely aligns with evaluations found elsewhere in the literature of causal discovery, so overall even in its current state it should be a useful addition to the literature.

**Additional Comments On Reviewer Discussion:**

There has been extensive discussion and both the reviewers and authors were deeply engaged. Key topics:
* Clarifications / notation: These have been largely resolved during the discussion
* Experiments: The authors provided additional experiments (and baselines) which were appreciated by the reviewers
* Downstream task (see weaknesses and paragraph below).

---

### Decision · Program_Chairs · 2025-01-22

Accept (Poster)